# Hydrodynamic tearing of bacteria on nanotips for sustainable water disinfection

Lu Peng ®[1,5], Haojie Zhu ®[2,5], Haobin Wang ®[3], Zhenbin Guo[2,4], Qianyuan Wu ®[1] ✉, Cheng Yang ®[2] ✉ & Hong-Ying Hu ®[1,3] ✉

Water disinfection is conventionally achieved by oxidation or irradiation, which is often associated with a high carbon footprint and the formation of toxic byproducts. Here, we describe a nano-structured material that is highly effective at killing bacteria in water through a hydrodynamic mechanism. The material consists of carbon-coated, sharp $Cu(OH)_2$ nanowires grown on a copper foam substrate. We show that mild water flow (e.g. driven from a storage tank) can efficiently tear up bacteria through a high dispersion force between the nanotip surface and the cell envelope. Bacterial cell rupture is due to tearing of the cell envelope rather than collisions. This mechanism produces rapid inactivation of bacteria in water, and achieved complete disinfection in a 30-day field test. Our approach exploits fluidic energy and does not require additional energy supply, thus offering an efficient and low-cost system that could potentially be incorporated in water treatment processes in wastewater facilities and rural communities.

The human race has fought against pathogenic microbes throughout history[1]. Waterborne pathogens have long been a threat to public health, and are associated with great pain and suffering[2]. The development of disinfection techniques including chlorination, ultraviolet radiation and ozonation has helped eliminate waterborne pathogens and improved the quality of life[3]. However, current disinfection practices rely on strong oxidants or harsh conditions[4,5], leading to a high carbon footprint and unpredictable health risks (e.g. carcinogenic byproducts[6,7] and microbial resistance[8,9]). Most of these technologies require a large-scale infrastructure and extensive maintenance, and therefore cannot be easily deployed in rural areas with inadequate electric power[10,11]. At present, billions of people worldwide still lack access to clean water and sanitation[12]. To provide universal access to safe and affordable drinking water, new disinfection processes that produce less secondary pollution and require less energy are urgently needed.

Recent advances in the mechano-bactericidal effects of nanomaterials provide a chemical-free approach for bacterial control[13–15]. It is generally believed that if enough mechanical force is exerted on a bacterium by surface contact, its cell wall can be penetrated[16]. However, bacteria have a natural resistance to mechanical shock from the environment[17], as reported by Suo et al., who showed that a bacterial cell remained viable after repeatedly puncturing it with a sharp atomic force microscopy (AFM) probe ($r \sim 35\,nm$)[18]. Previous studies have shown that mechano-bactericidal effects are more pronounced when bacteria were statically attached on the nanostructured surface to allow a sufficient disruption of cell integrity[19]. Incorporation of capillary force or surface tension at the air-liquid interface could help to achieve rapid cell deformation at the nanostructured surface, leading to its death[20]. Yet, this condition is not easily achieved in bulk water disinfection featured by a high throughput and a fluidic environment.

A fundamental characteristic of water is its fluidity, and using the mechanical energy in water flow to inactivate bacteria would be an ideal way to sustainably disinfect water. In a fluidic environment, the motion of bacteria is dominated by hydrodynamic forces and Brownian motion, which lead to random collisions during water flow[21,22]

[1]Shenzhen Key Laboratory of Ecological Remediation and Carbon Sequestration, Institute of Environment and Ecology, Tsinghua Shenzhen International Graduate School, Tsinghua University, Shenzhen, China. [2]Institute of Materials Research, Tsinghua Shenzhen International Graduate School, Tsinghua University, Shenzhen, China. [3]School of Environment, Tsinghua University, Beijing, China. [4]Institute of Semiconductor Manufacturing Research, Shenzhen University, Shenzhen, China. [5]These authors contributed equally: Lu Peng, Haojie Zhu. ✉e-mail: wu.qianyuan@sz.tsinghua.edu.cn; yang.cheng@sz.tsinghua.edu.cn; hyhu@tsinghua.edu.cn

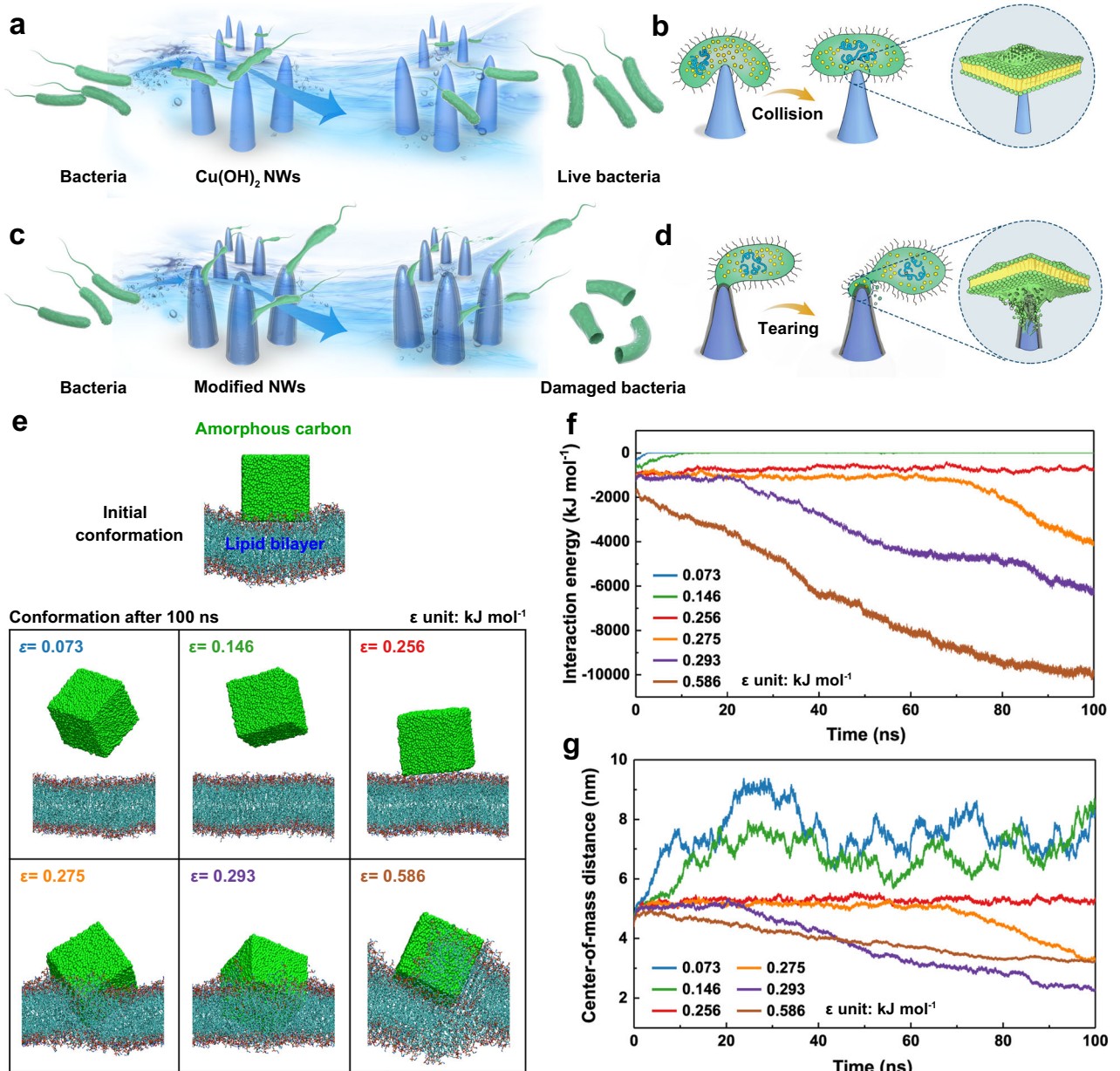

**Fig. 1 | General principles of the hydrodynamic-bactericidal mechanism.**
**a**, **b** Interaction between the bacteria and the nanostructured surface with a weak
London dispersion interaction in a fluidic environment. The water flow prompts the
bacterial cells to collide with the nanostructure, and the bacteria only encounter
resilient cell deformation. **c**, **d** The strong dispersion interaction between the
bacteria and the nanostructured surface allows cells to be trapped and torn up by
water flow. **a**–**d** are produced by Photoshop CC 2022. **e** Initial and final
configurations of the POPE lipid bilayer interacting with materials of different well
depth ($\varepsilon$) values during a 100-ns MD simulation. The material is shown in green
beads and the lipids in the membrane are shown in lines. Changes with time in the
interaction energy (**f**) and center-of-mass (COM) distance (**g**) between the tested
materials and the membrane. The interaction energy is in the form of van der Waals
forces. Source data are provided as a Source Data file.

(Fig. 1a). However, bacteria have evolved cell envelope to mechanically
resist external forces, which were up to 2–20 nN as obtained by
AFM[18,23]. Therefore, most cells experience resilient deformation with-
out any physiological structural damage when they collide, and this
does not significantly change even at a sharp and rigid nanostructured
surface (Fig. 1b). The rupture of bacteria during flow was only observed
when the flow was stopped so that the cells could be adhered on the
surface[24]. At present, there is no reported way of destroying bacteria in
a continuous flow condition using fluidic energy.

London dispersion force is a basic form of attractive interaction,
and it is vital to structural stability at the microscale[25,26]. It contributes
to the membrane integrity[27] and determines cellular functions such as
membrane permeability[28] and cell adhesion[29]. To destroy the

physiological structure of bacteria, dispersion interactions between
the contact surface and bacteria should not be ignored, as they play a
key role in transforming the kinetic energy from water flow to the cell
wall. At high flow speeds (e.g. a turbulent flow), the contact time is very
short, and thus a stable London dispersion interaction between bac-
teria and the contact surface is not reached. However, a mild fluidic
condition with a relatively low flow rate (e.g. a laminar flow) can only
deliver a small kinetic energy. Facing this dilemma, it is imperative to
set up a new force model incorporating dispersion interaction at the
microscale, so as to increase the efficiency of energy transfer from the
water flow to the cell envelope.

Here, we use a contact surface with nanotips which substantially
increases the stress delivered by the water flow to the cell envelope.

With this unique structure, we demonstrate a hydrodynamic-bactericidal mechanism which couples mild fluidic energy and London dispersion force between the nanotip surface and the cell envelope, leading to a dramatic bactericidal effect in water flow. We confirm that the stress produced by the hydrodynamic and dispersion forces is outward of the cell and overcomes the puncture resistance of the bacteria. Using this method, we inactivated >99.9999% of the bacteria in water and achieved continuous mechanical disinfection in a 30-day field test, demonstrating the potential of using environmental mechanical energy to destroy pathogenic bacteria.

## Results

### Illustration of hydrodynamic-bactericidal mechanism

The basic principles of the process are presented in Fig. 1c. We set up a model nanostructured surface with a strong dispersion interaction with bacteria, which enables an efficient energy transfer from the water flow to the cell envelope. When a bacterium collides with this nanostructure in flowing water, it is transiently trapped on the surface of the nanotips due to their strong attraction. The drag force of the flow stresses the contact area and therefore induces a considerable outward tension on the cell envelope, which is strong enough to overcome the puncture resistance of the bacterium, causing it to rupture and die (Fig. 1d).

To quantitatively investigate the London dispersion interaction, the well depth ($\varepsilon$) of the van der Waals (vdW) potential, which represents the energy of a system at the equilibrium state, was used to reflect intermolecular interactions[30]. The cell membrane was modeled with 1-palmitoyl-2-oleoyl-$sn$-glycero-3-phosphoethanolamine (POPE), which is the typical lipid molecule in the cell envelope[31]. Six different materials with different $\varepsilon$ values were placed close to a POPE lipid bilayer and showed distinctly different configurations after 100-ns free molecular dynamics (MD) simulations (Supplementary Movies 1–6). As shown in Fig. 1e, materials with $\varepsilon < 0.256$ kJ mol$^{-1}$ swung away from the lipid membrane while those with $\varepsilon > 0.256$ kJ mol$^{-1}$ strongly interacted with the lipid molecules and were inserted into the membrane.

To better understand these effects, we analyzed the physical interactions between the materials and the lipid membrane. Figure 1 f, g shows the changes with time in the interaction energy for different $\varepsilon$ values, together with the distances of the center-of-mass of these materials from the membrane. For materials with $\varepsilon < 0.256$ kJ mol$^{-1}$, a high-energy plateau (approximately 0 kJ mol$^{-1}$) indicated a weak attraction interaction in the system. A relatively constant energy value was found at $\varepsilon = 0.256$ kJ mol$^{-1}$ where the material was absorbed at the surface of the membrane without insertion. For $\varepsilon > 0.256$ kJ mol$^{-1}$, the vdW interaction energy decreased rapidly when the material was inserted into the lipid membrane, corresponding to a strong dispersion interaction between the material and the lipid molecules[32].

Although the POPE lipid bilayer is a simplified model for bacterial membrane, it is widely accepted and has been proved significant in analyzing molecular mechanism of bacteria-nanomaterial interactions[31,33,34]. The simulation of cell membranes with realistic components, including lipid, protein, and peptidoglycan, is the future direction, which is essential to unravel the behavior of a real cell membrane. However, due to the requirements of enhanced sampling algorithms and substantial data processing, there will be a continuing demand for simplified models containing few components[35].

Our theoretical simulations based on the POPE lipid bilayer model show an important finding that $\varepsilon > 0.256$ kJ mol$^{-1}$ is essential for strong attraction between the surface and the cell membrane. Based on these results, there may be a large number of materials with this property. For example, sp$^2$-carbon, which widely exists in nature and is chemically stable, has large numbers of delocalized electrons to produce a strong dispersion interaction with a bacterial membrane with $\varepsilon$ of 0.293 kJ mol$^{-1}$, and is an excellent candidate.

### A model nanostructured surface

We chose copper foam as the substrate for the production of the nanotip contact surface. This is an easily accessible material with a porous three-dimensional structure (Supplementary Fig. 1a) to allow water flow and cell collision. A high density of Cu(OH)$_2$ nanowires (Cu(OH)$_2$ NWs) with the diameter ~200 nm and length up to 5 μm was grown on the foam (Supplementary Fig. 1b, c), which provides numerous contact points on their sharp tips. Thermal treatment was used to coat these nanowires with a carbon layer to change the magnitude of the London dispersion force of the nanostructured surface. No obvious morphological change was observed after carbon coating and the sharp tips of the original Cu(OH)$_2$ NWs were well preserved (Supplementary Fig. 1d). X-ray diffraction (XRD) measurement was performed to investigate the crystalline structure of the modified NWs, which indicates that the Cu(OH)$_2$ phase was maintained after carbon coating (Supplementary Fig. 2). A slight change in surface hydrophilicity was observed (Supplementary Fig. 3), which confirms the coating of carbon on the modified NWs.

A transmission electron microscopy (TEM) image shows the morphology of an individual modified NW (Fig. 2a). Using an aberration-corrected TEM (ACTEM), a layer of amorphous carbon is seen evenly covering the surface of the nanowire with a thickness of about 15 nm (Fig. 2b). The surface carbon can be discriminated from the Cu substrate by their different image contrast in a bright-field scanning TEM (BF-STEM) and a high-angle annular dark-field scanning TEM (HAADF-STEM), as by elemental mapping (Fig. 2c). The chemical composition of the carbon layer was studied by Raman and X-ray photoelectron spectroscopy (XPS). The Raman spectra of the modified NWs shows a graphitic D-band at about 1580 cm$^{-1}$ (Fig. 2d), which is consistent with the main peak at 284.7 eV in XPS C 1$s$ spectra (Supplementary Fig. 4), demonstrating that the surface carbon is dominated by sp$^2$ C–C bonds.

We used atomic force microscopy (AFM) to confirm the attractive force between the amorphous carbon layer and the bacterial cell membrane (see Methods). An AFM tip was treated by the same coating method to modify the Cu(OH)$_2$ NWs (Supplementary Fig. 5). From Fig. 2e, we see that the cantilever's retraction was hindered as a result of adhesion between the carbon layer and the cell surface, while the original AFM tip exhibited no hysteresis in cell surface detachment. In this testing condition, the adhesion force between the carbon-coated AFM tip and the bacterium was measured to be 0.9 ± 0.5 nN (Supplementary Fig. 6), while there was a negligible adhesion force for the uncoated AFM tip (Supplementary Fig. 7). This measurement confirms that the amorphous carbon surface has a strong attraction to the bacterial cell due to the enhanced London dispersion force, which was verified in the above MD simulation.

### Bactericidal performance of modified NWs

In our tests, the copper foam has a filter-like porous geometry with an average pore size of 200 μm, which causes over 99.9999% of the bacteria to collide with a 3-mm thick membrane (Supplementary Fig. 8 and Supplementary Movie 7). To evaluate the bactericidal efficiency of the modified NWs in water flow, the Gram-negative bacterium *Escherichia coli* (*E. coli*) was used as the indicating microorganism and was suspended in sterilized water with a concentration of $10^6$–$10^7$ colony-forming units per milliliter (CFU mL$^{-1}$). We first compared the bactericidal performance of modified NWs with two control materials, Cu(OH)$_2$ NWs and modified Cu foam (Supplementary Fig. 9), in a flow-through cell at a flux of 2 m$^3$ h$^{-1}$ m$^{-2}$ (Supplementary Fig. 10). These control materials were used to exclude the contribution of the Cu substrate, the amorphous carbon itself or other related factors. The logarithmic removal efficiency was defined by $-\log(C/C_0)$, where $C$ and $C_0$ represent the live bacterial concentrations in the treated and untreated water samples respectively. As shown in Fig. 2f and Supplementary Table 1, the modified NWs achieved a superior disinfection

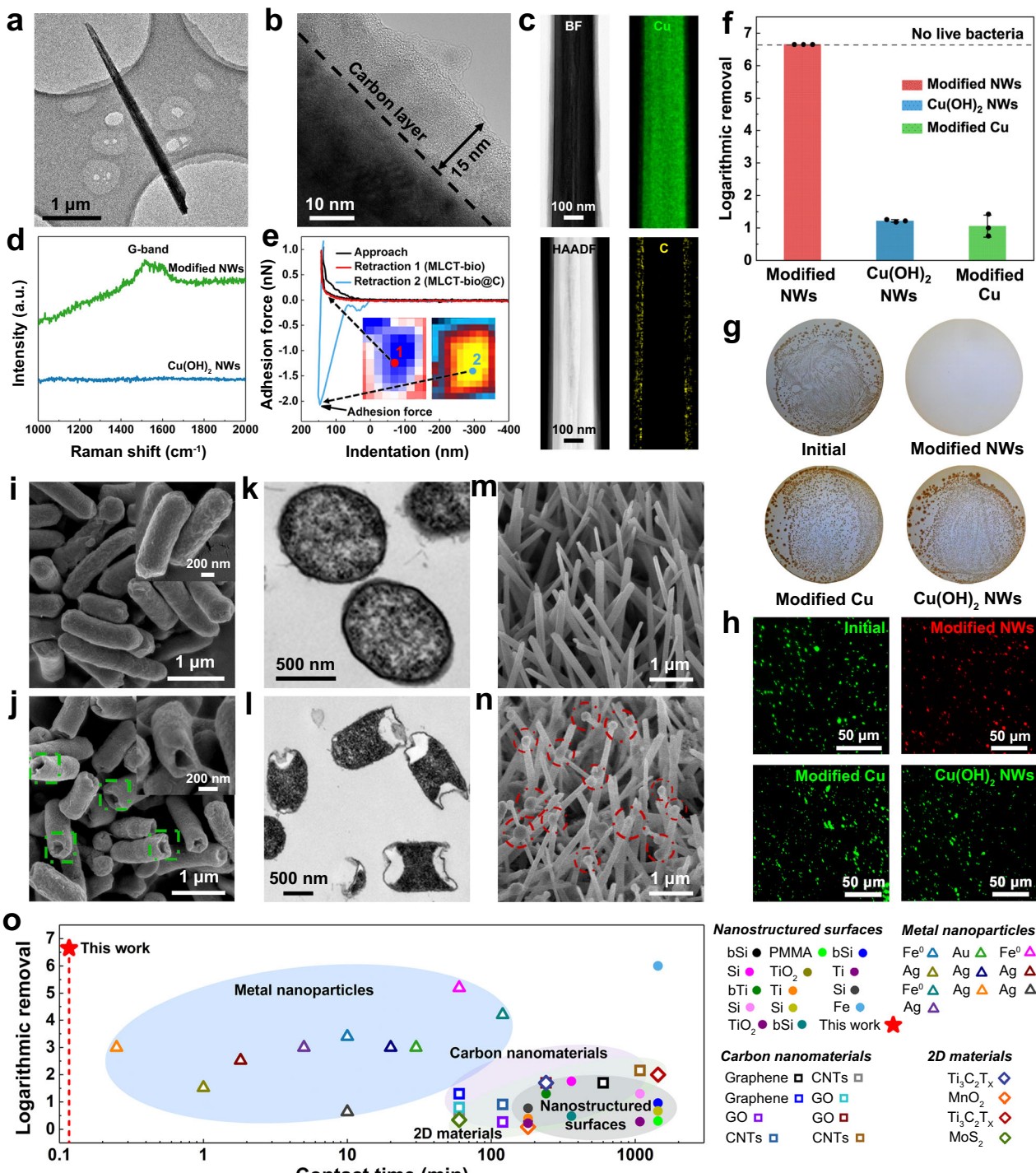

**Fig. 2 | Bactericidal performance analysis.** TEM (**a**) and ACTEM (**b**) images of the modified NWs. **c** AC-BF-STEM and AC-HAADF-STEM images of the modified NWs and corresponding elemental map. **d** Raman spectra of the Cu(OH)$_2$ NWs and the modified NWs. **e** Cell-tip adhesion analysis by AFM. The amorphous carbon coated AFM tip showed a distinctive hysteresis during retraction from the cell surface (blue line). **f** Bactericidal performance of the modified NWs, Cu(OH)$_2$ NWs, and modified Cu foam. Data in **f** are presented as mean ± SD with n = 3 independent experiments. **g** Cell-culture plates showing *E. coli* concentrations in the initial and treated water samples. **h** Fluorescence microscope images for *E. coli* in the initial and treated water samples (live cells are stained in green and dead cells are stained in red). SEM images showing morphologies of the initial *E. coli* (**i**) and *E. coli* treated by the modified NWs (**j**). Inset shows a single *E. coli* at a higher magnification. TEM images showing the ultrastructure of the initial *E. coli* (**k**) and *E. coli* treated by modified NWs (**l**). Morphologies of modified NWs before (**m**) and after disinfection (**n**). **o** Comparison of bactericidal efficiency between the modified NWs and comparable mechano-bactericidal activities. Observations using SEM or TEM (**a**–**c**, **i**–**n**) were repeated three times independently with similar results. Source data are provided as a Source Data file.

of *E. coli* by more than 99.9999% (>6 log removal). By contrast, the disinfection efficiency of the Cu(OH)$_2$ NWs and modified Cu foam were relatively low (~1 log), with large quantities of live bacteria remaining (Fig. 2g). *E. coli* viability was also assessed through a live/dead

fluorescence assay. Bacteria with intact cell membranes were stained with SYTO 9 (green), whereas nonviable bacteria with damaged membranes were stained with propidium iodide (red). It is clear from Fig. 2h that the bacteria treated by the modified NWs were stained red,

indicating severe membrane damage, while bacteria treated by the Cu(OH)$_2$ NWs and the modified Cu foam remained viable. These results indicate that the removal of bacteria in the flow was related to a combined effect between the nanotip structure and the surface carbon layer.

To obtain insight into the physiological structure changes of the bacteria after contact with the modified NWs, we examined the morphology of *E. coli* by scanning electron microscopy (SEM). The initial *E. coli* were rod-shaped with an intact cell membrane (Fig. 2i); however, in the effluent, they were severely damaged and had many pores (Fig. 2j). The rupture sizes of the treated bacteria were measured to be 100–200 nm from the SEM image, with an average value of 122 ± 32 nm (Supplementary Fig. 11). The TEM analysis (Fig. 2k, l) also indicates that the cell envelope was ruptured, leading to partial leakage of the cytoplasmic contents. To confirm this cell damage, the *E. coli* cells treated by the modified NWs were immediately imaged in water by structured-illumination microscopy with the presence of lipophilic cyanine dye DiO (green) and propidium iodide (red). After flowing through the modified NWs, the *E. coli* cells had a compromised membrane that allowed the entrance of red propidium iodide (Supplementary Fig. 12). Furthermore, a number of fragments (red dashed circles) were attached to the modified NWs after a continuous disinfection test (Fig. 2m, n). As can be seen from a higher-magnification SEM image (Supplementary Fig. 13a, b), the dimension of the fragments was in the range of 100–200 nm, which agrees with the rupture sizes of the cell envelope. Considering that the bactericidal test was conducted in deionized (DI) water, these fragments are likely to be bacterial debris that were torn from the cell bodies while the damaged cells were flushed away in the flow. In contrast, the unmodified Cu(OH)$_2$ NWs did not show any debris on the surface after continuous disinfection (Supplementary Fig. 13c, d), which suggests that the carbon layer induced a different interaction mode between the bacteria and the nanowires due to a higher London dispersion force, causing the tearing of the bacteria during flow.

We further confirmed the negligible contributions of other potential mechanisms to the removal of bacteria, including adsorption, oxidative stress, and toxicity of the released copper ion (Cu$^{2+}$). The optical density of the effluent water was comparable to the influent water (Supplementary Fig. 14), meaning that the density of the bacterial cells was unchanged in the effluent and the bacteria were not removed by adsorption. In addition, the elevation of intracellular reactive oxygen species (ROS) was not observed in the bacteria treated by the modified NWs (Supplementary Fig. 15). Therefore, the influence of oxidative stress is negligible. Note that the effluent Cu$^{2+}$ concentration (0.3–0.5 mg L$^{-1}$) was far below the guideline of World Health Organization for safe drinking water (2 mg L$^{-1}$)[36], and the contribution of released Cu$^{2+}$ ions to bacterial removal was limited (Supplementary Fig. 16). From these observations, we conclude that mechanical destruction was the major cause of bacterial inactivation by the modified NWs.

This bactericidal process is different from the reported mechano-bactericidal activities in previous studies (Fig. 2o and Supplementary Table 2). As is shown in Fig. 2o, there are large variations in the contact time and the bactericidal efficiencies among different types of nanomaterials. Typically, previously reported mechano-bactericidal activities are based on a surface-contact mechanism, in which bacterial cells are deformed during static contact with the sharp nanostructures[37,38]. This mostly requires a long contact time (up to hours) to deliver sufficient stress beyond the elastic limit of the cell envelope[14,39–41]. Metal nanoparticles show higher bactericidal activities due to their toxicity and induction of oxidative stress after translocating into the cell[42,43], apart from their mechano-bactericidal behaviors. Here, we report a fluidic energy triggered tearing mechanism, in which bacteria are torn apart by an instantaneous contact with nanotips during flow. The coupling of the hydrodynamic force and London dispersion

interaction between the nanotip surface and the cell envelope achieved >99.9999% inactivation of the bacteria within a short contact time (e.g. 7 s), which is, to the best of our knowledge, the first observation of effective mechanical disinfection in bulk water.

## From puncturing to tearing: discussions of the cell rupture mechanism

The design of a filter-like porous flow-through unit allows effective cell collision with the surface (>99.9999%) (Supplementary Fig. 8 and Supplementary Movie 7) and the formation of nanotips reduces the contact area with bacteria, leading to an enlarged stress on the cell envelope. During an instantaneous collision with a nanotip, the collision energy deforms the bacteria (Fig. 1a, b). While the high London dispersion force at the surface of the modified NWs creates a transient attachment of the bacteria to the nanotips. Subsequent flow produces tearing stress on the cell envelope (Fig. 1c, d). To determine whether the bacteria were ruptured by puncturing or tearing, we developed a biophysical model to analyze the forces exerted on the bacterial cell during flow.

Before this we measured the mechanical properties of the *E. coli* cell. Measurements of mechanical properties of the bacterial cell are sensitive to the experimental conditions, and probing live cells under physiological conditions is therefore an ideal way to investigate the biophysical mechanism of bacteria[17,44]. Here we used AFM to directly probe live *E. coli* in a liquid environment. We followed the protocol of a puncture experiment[18] and obtained force versus displacement curves of bacterial cells (see Methods). A typical puncture curve is shown in Fig. 3a, and more information is provided in Supplementary Fig. 17. The Young's modulus of the bacterial cell is determined from the initial part of the loading force versus cell indentation (see inset in Fig. 3a). By fitting the data with the classic Sneddon and Hertz Model (Supplementary Table 3), we obtained an average Young's modulus of 0.5 MPa, which is comparable to the reported results measured in liquid environment (Supplementary Table 4). The critical point at which the AFM tip broke into the cell wall appeared at a maximum cell indentation of 85 nm (Fig. 3a). We calculated the stress distribution profile of the bacterial cell envelope at this critical point using the obtained Young's modulus (Fig. 3b). The maximum pressure appeared at the edge of the contact area, with a value of 0.05 MPa, which is considered the critical stress required to rupture the cell.

To investigate the detailed process during the contact between the bacteria and the surface during flow, we further analyzed the bacterial movement near the surface nanostructure by a Brownian dynamics and computational fluid dynamics method using a set of cylindrical tips perpendicular to the horizontal surface to present the nanotips on the copper foam. The flow rate in the main flow was set to 5.5 × 10$^{-4}$ m s$^{-1}$ corresponding to a flow rate of 2.7 mL min$^{-1}$ in the experimental conditions. The velocity of the flow near the contact surface was calculated and is shown in Fig. 3c, where the flow rate was much lower than that in the main flow, with a value of 5 × 10$^{-5}$ m s$^{-1}$. Then, movement of the *E. coli* cell in the flow field was simulated by importing the calculated flow information into a Brownian dynamics equation[45]. We simulated 100 cells in the defined flow field and obtained eight different types of contact between a bacterium and nanotips (Fig. 3d and Supplementary Fig. 18). Because *E. coli* is rod-shaped, the contact with the nanotips can be either at the end (six types) or the middle (two types), and the possibility of end-contact (57%) is higher than that of middle-contact (43%).

Finally, we modeled the stress distribution profile of a cell membrane based on its interaction with the nanotips. We first considered puncturing during the collision process (Supplementary Movies 8 and 9). During collision, we assumed the work done by the tip to a bacterium equaled the loss of kinetic energy. The maximum stresses for the end-contact and middle-contact types were calculated to be 2.61 × 10$^{-4}$ and 4.54 × 10$^{-4}$ MPa, respectively (Fig. 3e, f), which are two orders

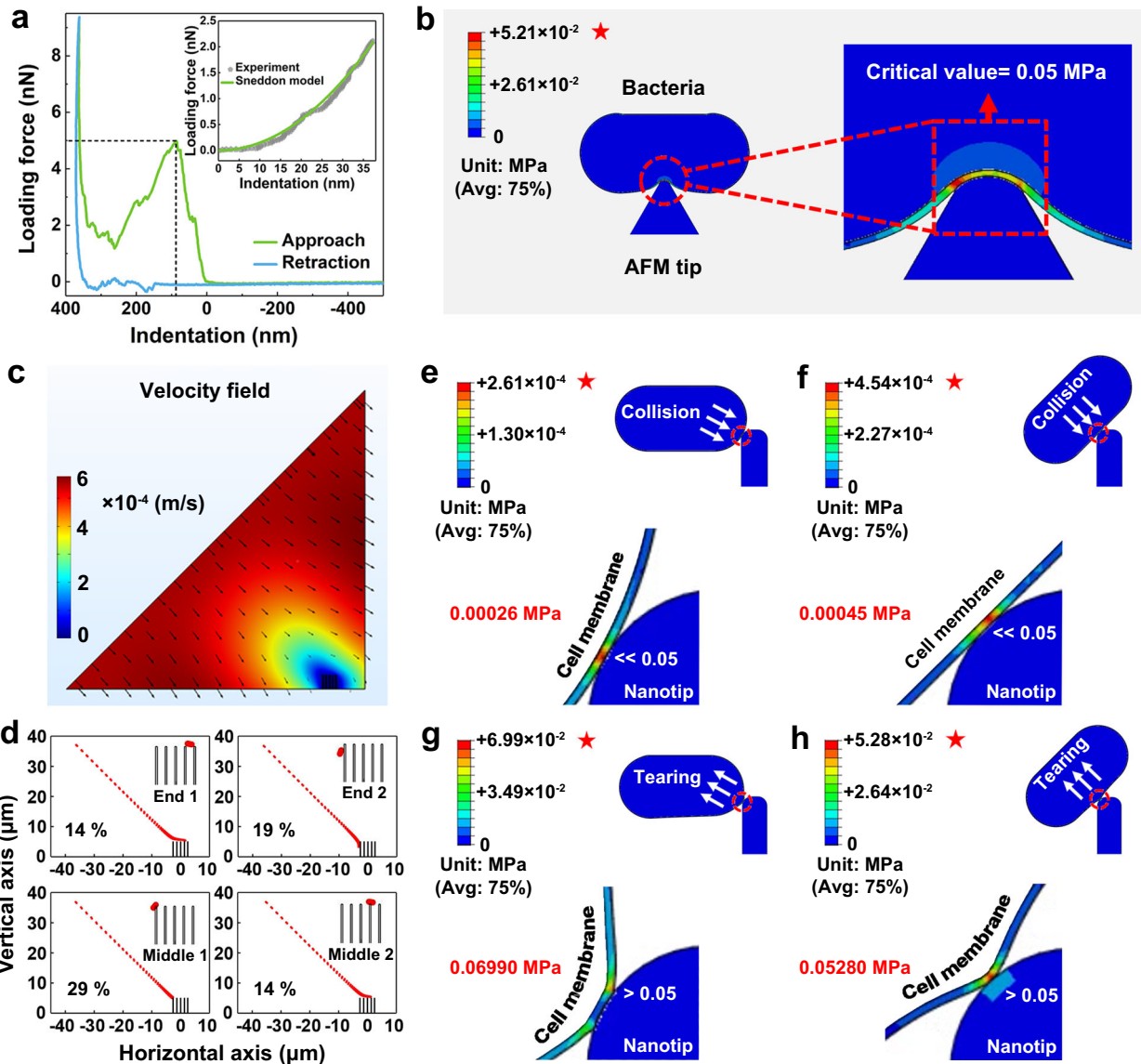

**Fig. 3 | Cell rupture by the hydrodynamic-bactericidal effect. a** A typical puncturing curve of *E. coli* obtained with AFM. The inset shows the fit of the data in the early part of indentation using the Sneddon model. **b** Finite element method simulation of the cell penetration process of the AFM tip. **c** The calculated velocity field of the simulated area. A right triangle domain was selected for computation, in which the hypotenuse was defined as the inlet boundary, while the other two sides were defined as the outlet boundaries. **d** Possibilities of the four types of contact between the bacteria and the nanotips. Stress distribution profiles of the cell membrane during collision process by end-contact (**e**) and middle-contact (**f**). Stress distribution profiles of the cell membrane during tearing process by end-contact (**g**) and middle-contact (**h**). The maximum stresses exerted at each contact form are denoted in red, and are compared with the critical stress (0.05 MPa). Source data are provided as a Source Data file.

of magnitude lower than the critical value (0.05 MPa). Such data suggests that the collision process cannot mechanically rupture a bacterium, and explains the poor bactericidal performance of the surface of the Cu(OH)$_2$ NWs (Fig. 2f), where only the puncturing effect exists. When bacteria collided with the surface of modified NWs, the water flow combined with the higher London dispersion interaction to exert a tearing effect (Supplementary Movies 10 and 11). The maximum outward stresses in the end-contact and middle-contact form were calculated to be $6.99 \times 10^{-2}$ and $5.28 \times 10^{-2}$ MPa, respectively (Fig. 3g, h), which exceed the rupture stress of the bacteria. In our simulation, the drag force of the flow was estimated by the minimum flow rate and the random torque of the flow was ignored, which theoretically induces rotation of the cell body and exerts extra tension for membrane deformation[22,46]. Hence, the stress in the tearing process was sufficient to rupture the bacterial cell. The numeric simulation results are in good agreement with the experimental results of Cu(OH)$_2$ NWs and

modified NWs, which allow us to demonstrate that the tearing generated by the hydrodynamic and dispersion forces is the true reason for the cell rupture rather than hydrodynamic/Brownian collisions.

## Practical disinfection applications

Based on the hydrodynamic-bactericidal mechanism, we have produced a model disinfection system (Fig. 4a). The modified NWs were placed in a chamber, and the contaminated water flowed into the chamber for disinfection. During a short time in the chamber, the bacteria suffered destructive mechanical damage and lost cell integrity by contact with the modified NWs. This prototyped continuous water disinfection system can be integrated into municipal water pipelines and be easily scaled up by stacking the chamber units with the modified nanotips in series.

We first evaluated the influence of flow rate on the performance of this novel disinfection system (Supplementary Fig. 19). Limited

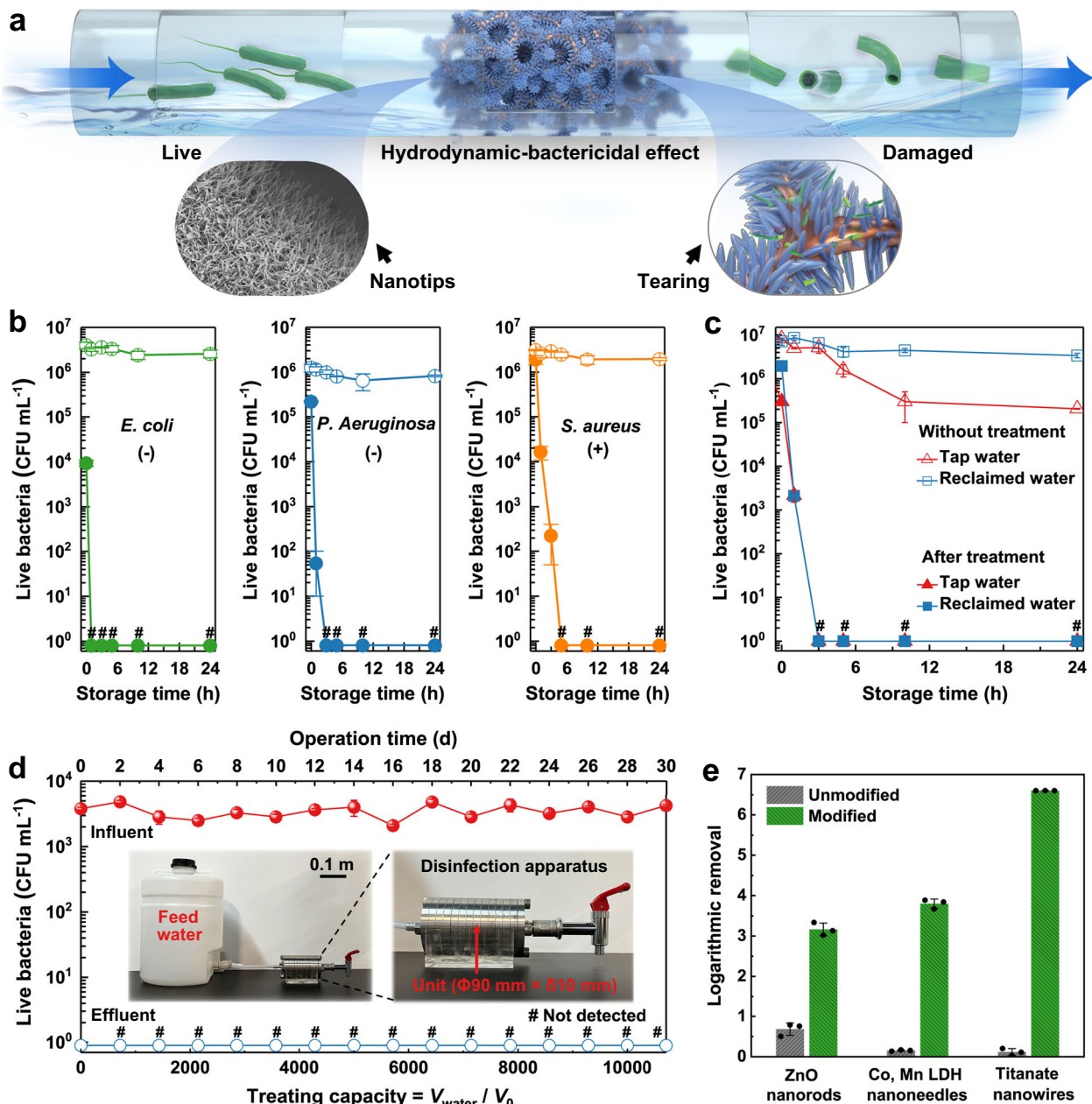

**Fig. 4 | Practical disinfection applications. a** Schematic of the hydrodynamic-bactericidal mechanism for practical water disinfection (produced by Photoshop CC 2022). **b** Bacterial storage experiments under visible light for 24 h using three representative bacteria including Gram-negative *E. coli*, *P. aeruginosa*, and Gram-positive *S. aureus*. The lines with hollow circles represent bacteria without treatment and the lines with solid circles represent bacteria after treatment. **c** Disinfection in real water samples including tap water and reclaimed water. **d** Influent and effluent bacterial concentrations during a 30-day field test. Inset shows optical images of the model flow-through disinfection apparatus. Each unit has an outer diameter of 90 mm and a thickness of 10 mm. The treating capacity is defined as the ratio of actual treated volume ($V_{water}$) to the effective volume of the chambers ($V_0$). **e** Bactericidal performance of the unmodified and modified ZnO nanorods, Co, Mn-LDH nanoneedles and titanate nanowires. Data are presented as mean ± SD with $n = 3$ independent experiments for (**b**, **c**, **e**), and $n = 3$ independent measurements for (**d**). # indicates below detection limit 1 CFU mL$^{-1}$. Source data are provided as a Source Data file.

bactericidal activity (-1.4 log) was achieved in the static condition, indicating that the significance of water flow to the rupture of the bacteria. In the flow condition, the modified NWs showed remarkable inactivation of *E. coli* over a range of flux (0.5–6 m$^3$ h$^{-1}$ m$^{-2}$). Complete disinfection (>6 log removal) was observed at fluxes of 0.5 and 2 m$^3$ h$^{-1}$ m$^{-2}$, which are the commonly-adopted fluxes in filtration modules[47,48]. Yet there is a decrease in the inactivation efficiency at a higher flux of 6 m$^3$ h$^{-1}$ m$^{-2}$. This is because at such a high flux, the contact time of the bacteria within the materials was decreased, leading to a substantial decrease in the contact possibility between the bacteria and nanotips[49].

To confirm the robustness of the disinfection effect, we evaluated the persistence of bacterial inactivation caused by this hydrodynamic-bactericidal mechanism. We disinfected three representative Gram-negative and Gram-positive bacteria and assessed their viability using a 24-h storage experiment, including *E. coli*, *Pseudomonas aeruginosa* (*P. aeruginosa*), and *Staphylococcus aureus* (*S. aureus*). As shown in Fig. 4b, the three types of bacteria completely lost their viability after disinfection (>6 log inactivation), and no regrowth or reactivation was observed during storage. A significant inactivation was also observed for Gram-positive *S. aureus*. Similar to *E. coli*, the cell envelope of the *S. aureus* cell was damaged with holes on the surface after flowing

through the modified NWs (Supplementary Fig. 20). Yet there was only a slight leakage of cytoplasm, suggesting that *S. aureus* is more resistant to mechanical damage compared with Gram-negative *E. coli*, due to a thicker peptidoglycan cell wall[50].

We tested the disinfection performance in real water samples using *E. coli* as the indicating microorganism (Fig. 4c). The characteristics of the tap water and reclaimed water are shown in Supplementary Table 5. After flowing through the modified NWs, we observed a rapid decrease of the live bacterial concentration in both water samples. The treated bacteria did not reactivate during the 24-h storage under visible light illumination. Because of mechanical destruction, the treated cells could not recover with the extension of membrane damage and loss of cytoplasm[51], which avoids the risks of bacterial regrowth in conventional disinfection processes such as the ultraviolet radiation[52].

A flow-through disinfection apparatus was constructed to test the long-term performance of this model system (Fig. 4d inset). A series of stainless-steel chamber units was connected for evaluation during a 30-day continuous disinfection. A bacterial solution containing $10^3$–$10^4$ CFU mL$^{-1}$ *E. coli* was used as the feed water, simulating the typical bacterial concentrations found in water purification system[53]. During a 30-day continuous operation, no live bacteria were detected in the effluent (Fig. 4d), which corresponds to a treating capacity of over 10,000 times the effective volume of the chambers. The morphology of modified NWs after 30-day operation was evaluated by SEM. The bacterial debris only accumulated in the first unit (Supplementary Fig. 21a, b), while in the following units, sparse debris was found and the morphology of the modified NWs was well preserved after 30-day flushing (Supplementary Fig. 21c–f). It was estimated that this flow-through disinfection apparatus with a chamber volume of 1 L can support the daily water consumption of an adult for over ten years (see Methods), showing its prospects for broad applications.

The hydrodynamic-bactericidal mechanism is solely a physical process in which the surface London dispersion interaction is the critical factor regardless of the type of surface with nanotip characteristics. This was shown by using three different nanomaterials, including ZnO nanorods, Co, Mn-layered double hydroxides (LDH) nanoneedles and titanate nanowires, in which the same carbon coating was used for all three materials (see Methods). Disinfection results show that only a weak bactericidal activity was observed in their original forms (Fig. 4e). After surface modification, the three nanostructured surfaces showed significant bacterial inactivation of >99.9%. The difference in the bacterial killing efficiency of these nanomaterials is associated with the differences in surface geometry (Supplementary Figs. 22–24). The diameter, height, and density of these surface patterns affect the contact between the bacteria and the nanotips[54], leading to changes in the local force distribution. For instance, the carbon-coated ZnO nanorods showed lower bactericidal efficiency than the carbon-coated Co, Mn-LDH nanoneedles and carbon-coated titanate nanowires. This is because ZnO nanorods have larger diameters and a lower tip density (Supplementary Table 6). Typically, nanostructures with a blunt feature are supposed to deliver less mechanical stress[41], due to an enlarged contact area. Besides, a lower tip density of ZnO nanorods also reduces the possibility of bacterial contact with the nanotips during flow, which negatively impacts their bactericidal performance. Nonetheless, these results indicate that the carbon coating treatment can increase the bactericidal performance of different nanomaterials by at least three orders of magnitude, confirming that the London dispersion interaction is the critical factor to determine the bactericidal efficiency of the nanotips in water.

## Discussion
In this study, we demonstrate a hydrodynamic-bactericidal mechanism which couples the mild fluidic energy and the London dispersion interaction between the nanotip surface and the cell envelope, leading to superior mechanical inactivation of bacteria in water. Although the applicability of this method was verified on different nanotip surfaces, we have yet to fully identify the influence of nanostructure geometry on the rupture of the bacteria, as the formed nanotip structures obtained by chemical methods cannot be ensured all geometrically identical. A thorough study of the effects of geometrical parameters requires precisely-controlled nanofabrication. We believe the advancement in nanofabrication techniques such as reactive ion etching or deep UV lithography may help to provide more precise analysis on the role of nanostructure geometry in the future[55].

Our method is effective against Gram-positive *S. aureus*, yet we observed the rupture of *E. coli* is more vigorous than *S. aureus* based on the morphology of the bacteria after disinfection. The influence of bacterial species is a complex issue. For instance, the difference in bacterial shape (e.g. rod shape or coccus) can affect the stress distribution profile when interacting with the nanotips[56]. In addition, the cell envelope composition not only governs the cell stiffness but also influences the level of London dispersion interaction between the bacteria and the nanotips. A thorough study incorporating the above issues should be carried out to give guidance for the design of a more reliable disinfection device. Besides, the effectiveness of this method towards other types of waterborne pathogens need to be studied, such as viruses, fungi and protozoa. As viruses possess a noncellular structure with a much smaller size, there is potential limitation of the current system for the inactivation of viruses. The fine adjustment of nanotip geometry is thus important to produce nanotips comparable to the size of viruses.

Furthermore, studies under real-world conditions are required. Colloids, particles, dissolved organic matter, and ions coexist with pathogens in realistic water treatment conditions[57]. These substances may absorb to the modified NWs, shield the effective sites on the nanotip surface, and possibly change the level of dispersion interaction with the bacteria. Exploring the effects of these substances on the killing efficiency of the pathogens is therefore of great importance for its broad applications. For practical water treatment, the modified NWs can be combined with other conventional water treatment processes. For example, the influent water can be pretreated by an ultra-filtration module to remove most of the impurities[58].

Nevertheless, this study reports a methodology on exploiting fluidic energy to destroy pathogenic bacteria for the first time, which provides implications for the development of chemical-free disinfection technology to address global challenges in environment and healthcare. A superior inactivation of >99.9999% bacteria was achieved by simply flowing the water through the device. That is, besides the nanotip surface, flow of contaminated water is the only requirement to attain disinfection, which avoids toxic chemical byproducts and additional energy input. As a result, drinking water, municipal water and wastewater facilities as well as communities living in rural areas may benefit from this method of obtaining safe and clean water. It may also shed light on the development of future pathogenic control in other fields.

## Methods
### Materials and reagents
Sodium hydroxide (NaOH), ammonium persulphate ($(NH_4)_2S_2O_8$), glucose ($C_6H_{12}O_6$), and copper sulfate pentahydrate ($CuSO_4 \cdot 5H_2O$) were purchased from Shanghai Macklin Biochemical Co., Ltd., China. Hydrochloric acid, nitric acid, ethanol, and tert-butyl alcohol were purchased from Sinopharm Chemical Reagent Co., Ltd., China. Zinc nitrate hexahydrate ($Zn(NO_3)_2 \cdot 6H_2O$), hexamethylenetetramine ($C_6H_{12}N_4$), cobalt nitrate hexahydrate ($Co(NO_3)_2 \cdot 6H_2O$), manganese chloride tetrahydrate ($MnCl_2 \cdot 4H_2O$), urea ($CO(NH_2)_2$) and ammonium fluoride ($NH_4F$) were purchased from Shanghai Aladdin Biochemical Technology Co., Ltd., China. Ammonia was purchased from Shanghai Titan Scientific Co., Ltd., China. Copper foam, nickel foam and titanium

foam were obtained from Kunshan Guangjiayuan New Materials Co., Ltd., China. Nutrient broth and nutrient agar media were supplied by Qingdao Hope Bio-Technology Co., Ltd., China. *E. coli* (CGMCC 1.3373), *P. aeruginosa* (CGMCC 1.12483), and *S. aureus* (CGMCC 1.12409) were obtained from China General Microbiological Culture Collection Center (CGMCC). AFM probe (MLCT-bio) was purchased from Bruker. Poly-L-lysine (0.01 wt%) was purchased from Sigma-Aldrich. LIVE/DEAD BacLight Bacterial Viability Kit (L7007) and Vybrant DiO cell-labeling solution (V22886) were obtained from Invitrogen, USA. Reactive Oxygen Species Assay Kit (NO. S0033) was purchased from Beyotime Biotechnology Co. Ltd., China. Deionized (DI) water was produced by Milli-Q Water System (Millipore, USA) and all the solutions were prepared by DI water unless otherwise mentioned.

### Fabrication of $Cu(OH)_2$ NWs, modified NWs and modified Cu foam

$Cu(OH)_2$ NWs were synthesized on copper foam using chemical oxidation[59]. The copper foam with a thickness of 2 mm and an average pore size of 200 μm was cut into $3 \times 4$ cm$^2$ pieces and sequentially washed with ethanol, hydrochloric acid, and DI water to remove surface impurities. The cleaned copper foam was then immersed in 150 mL of an aqueous solution containing 2.5 M NaOH and 0.1 M $(NH_4)_2S_2O_8$ for 20 min at 4 °C to produce the $Cu(OH)_2$ NWs. It was removed from the solution, rinsed with DI water and dried in a vacuum oven.

The modified NWs were prepared by a simple thermal treatment. The $Cu(OH)_2$ NWs were placed downwind of a tube furnace with glucose in the heating zone, which was used as the carbon precursor and pyrolyzed at 550 °C in an Ar atmosphere for 2 h. The evaporated carbon settled on the surface of the $Cu(OH)_2$ NWs to form the modified NWs. To prepare the modified Cu foam, the cleaned copper foam was subjected to the same treatment.

### Material characterization

The morphologies of the fabricated materials were analyzed by scanning electron microscopy (SEM, HITACHI SU8010) and transmission electron microscopy (TEM, FEI Tecnai G2 spirit). The ultrastructure of the modified NWs was examined by aberration-corrected TEM (ACTEM, JEM-ARM300) using the bright-field scanning TEM (BF-STEM) and high-angle annular dark-field scanning TEM (HAADF-STEM) modes. The crystal structures of the samples were studied by X-ray diffraction (XRD, D8 Advance). The wettability of the samples was investigated by a contact angle measuring instrument (KRUSS DSA30). The chemical compositions of the samples were analyzed by X-ray photoelectron spectroscopy (XPS, PHI 5000 VersaProbe II) and Raman spectroscopy (Horiba LabRAM HR800).

### Bactericidal test

*E. coli* was used as a model pathogen to evaluate the bactericidal performance of the sample materials. Pure *E. coli* was cultured in nutrient broth at 37 °C with shaking at 150 rpm for 12 h to achieve a concentration of $10^9–10^{10}$ CFU mL$^{-1}$. The composition of the culture media is listed in Supplementary Table 7. The cultured bacteria were harvested by centrifugation and washed twice with sterile DI water. The prepared *E. coli* suspension was diluted to sterile DI water to obtain a concentration of $10^6–10^7$ CFU mL$^{-1}$.

Bactericidal tests were conducted in a flow-through Plexiglas cell with two pieces of prepared cupper foam placed inside (Supplementary Fig. 10). The copper foam was 2-mm thick with an effective filtration area of 78.5 mm$^2$. The flow rate of the water sample was fixed at 2.7 mL min$^{-1}$ by a peristaltic pump, which corresponds to a contact time of 7 s and a flux of about 2 m$^3$ h$^{-1}$ m$^{-2}$. Before the bactericidal test, the copper foam was washed by pure water in the flow-through cell for two hours to remove surface impurities. The fresh bacterial solution ($10^6–10^7$ CFU mL$^{-1}$) was used as the influent water and flowed into the

cell. During the test, the influent water was mixed by a magnetic stirrer. Normally, the bactericidal tests were completed within one hour, and thus about 160-mL water was treated for each test. For each type of material, three sets of flow-through cells were set up and operated independently. The bactericidal efficiency for each type of material was counted based on the three replicates.

The live bacterial concentrations in the influent and effluent water were measured using a standard plate count method. The time to spread the bacterial solution on the plate for each sample was normalized to one hour after sampling. Each water sample was diluted serially (1:10, 1:100, 1:1000 and 1:10,000), and spread onto a sterile Petri plate (in triplicate for each dilution), in which cooled and molten nutrient agar medium had been added. Following 24-h incubation at 37 °C, the number of the bacterial colonies formed on the plates was counted, and the concentration of bacteria in the original water sample was obtained by multiplying the number of colonies obtained per plate by the dilution factor.

The inactivation performance was evaluated by logarithmic removal efficiency, which was defined by $-\log(C/C_0)$, where $C$ and $C_0$ represent bacterial concentrations in the influent and effluent water obtained by plate count. When no colonies were formed on the plates, including original and diluted water samples, the bacteria in the water sample were considered as completely inactivated and the logarithmic removal efficiency was calculated by $\log(C_0)$.

### Live/dead viability assay

A LIVE/DEAD BacLight Bacterial Viability Kit was used to test the viability of the bacteria. The two dye components provided with the kit were mixed to achieve a concentration of 1.67 mM for SYTO 9 and 10 mM for propidium iodide, which provides good live/dead discrimination. 3 μL of dye mixture was immediately added to 1 mL of the bacterial sample, mixed thoroughly and incubated at room temperature in the dark for 15 min. The bacterial sample was then filtered through a black polycarbonate membrane (diameter 25 mm, pore size, 0.22 μm, Millipore, USA) and observed under a fluorescence microscope (Nikon, ECLIPSE Ni-U).

### Bacterial sample preparation for SEM

The morphology of the bacteria was investigated by SEM. The bacteria samples before and after treatment were harvested by centrifugation at $9700 \times g$ for 5 min and fixed with 2.5% glutaraldehyde at 4 °C for 12 h. Next, the bacterial samples were rinsed with water and dehydrated for 15 min with a series of ethanol/water solutions with increasing ethanol content (30%, 50%, 70%, 90%, 100%). The ethanol was then displaced in a series of tert-butyl alcohol content (40%, 60%, 80%, 100%) solutions. Finally, the turt-butyl alcohol in the sample was removed by freeze-drying.

### Bacterial sample preparation for TEM

The ultrastructure of the bacteria was assessed by TEM. The bacterial samples were first harvested by centrifugation and were fixed with 2.5% glutaraldehyde at 4 °C for 12 h. After being washed with water three times and postfixed with 1% osmium tetroxide at 4 °C for 1 h, the samples were rinsed again and dehydrated for 15 min with a series of ethanol/water solutions with increasing ethanol content (30%, 50%, 70%, 90%, 100%). The samples were then infiltrated with resin and cured overnight at 60 °C to form resin blocks. The resin blocks were sectioned and picked up on a TEM grid. Finally, the samples on the grid were stained by a 3% aqueous solution of uranyl acetate and 4% lead citrate solution for 10 min.

### Bacterial sample preparation for AFM

*E. coli* stock solution with a concentration of $10^9–10^{10}$ CFU mL$^{-1}$ was prepared as discussed earlier. A glass slide coated with poly-L-lysine (Sigma, 0.01 wt%) was incubated in a 100 μL bacterial solution for

20 min and was then incubated in sterile DI water at least three times to remove the unattached cells. The immobilized bacteria were used immediately for AFM experiments and placed in water during the test to prevent cell dehydration.

## Cell-tip adhesion measured by AFM

To prepare the carbon-coated AFM probe, an AFM tip (MLCT-bio, Bruker) was treated by the same method used for the modification of the Cu(OH)$_2$ NWs. It was washed with ethanol and DI water to remove surface impurities, and dried in air before use. The morphologies of the original and the carbon-coated MLCT-bio tips are shown in Supplementary Fig. 5.

The adhesion force between the AFM tip and bacterial membrane was analyzed in the force-volume mode by an Asylum AFM (MFP-3D-SA) according to the literature[18]. The cantilever deflection sensitivity was first calibrated in DI water on the surface of a clean glass slide (not covered with bacteria). The tip was then aligned at the center of an area where *E. coli* cells were evenly distributed under inverted optical microscope. Force versus deflection curves were acquired from a $7 \times 7$ μm$^2$ area which was divided into $32 \times 32$ grids. The cantilever force constant was about 0.065 N m$^{-1}$, and the maximum loading force was set at 1 nN. The experiments were performed at a frequency of 0.5 Hz, with the cantilever velocity around 1.98 μm s$^{-1}$. The tip with or without carbon coating was brought into contact with *E. coli* and the adhesion force was displayed on the retract curve.

## Cell mechanics measured by AFM

Mechanical measurements of *E. coli* cell were carried out by a puncture test, which were performed in the force-volume mode as mentioned above. An MLCT-bio tip was used, and the tip radius was measured to be 40 nm using SEM. The cantilever force constant was about 0.046 N m$^{-1}$, and the maximum loading force was set at 10 nN to allow cell penetration. The experiments were performed at a frequency of 1 Hz, with the cantilever velocity around 1.98 μm s$^{-1}$. The Young's modulus ($E$) of the *E. coli* sample was obtained by fitting the indentation curve obtained from the approach curve using a Sneddon or Hertzian model[60].

## Structured-illumination microscopy (SIM)

The *E. coli* samples were stained with propidium iodide (Invitrogen, USA) for 10 min, and then mixed with a commercial lipophilic cyanine dye DiO (Vybrant cell labeling solution, Invitrogen, USA) for another 5 min. The DiO stains bacterial cell membrane while propidium iodide binds only to the DNA of cells with a compromised membrane. The bacterial solution was transferred to a glass bottom dish and immediately imaged using the Nikon N-SIM apparatus with a 100× oil objective and laser excitation for DiO (488 nm), propidium iodide (561 nm). High-resolution images were acquired by 3D-SIM and reconstructed by slice 3D-SIM.

## Optical density measurement of the influent and effluent water

*E. coli* suspension with a concentration of $4 \times 10^6$ CFU mL$^{-1}$ (determined by plate count) was used as the influent water. Before the test, the modified NWs were washed by pure water in the flow-through cell for two hours to remove surface impurities. Effluent water was collected at different flow rates (0.5, 2 and 6 m$^3$ h$^{-1}$ m$^{-2}$). Optical density data of the influent and effluent water were measured at a wavelength of 600 nm and an optical pathlength of 5 cm using a HACH spectrophotometer (DR3900). DI water was used as the blank. The effluent samples of DI water were collected at the same flow rates to exclude the potential influence of released particles on optical density.

## Measurement of reactive oxygen species (ROS)

The intracellular levels of ROS in bacteria were determined by a Reactive Oxygen Species Assay Kit (Beyotime, No. S0033) containing a fluorescent probe DCFH-DA and a Rosup reagent. Fresh bacterial suspension ($10^6$–$10^7$) was first treated with 100 μM DCFH-DA in the dark for 1 h at 37 °C, which was then washed three times with sterile DI water to remove the residual DCFH-DA[61]. The bacterial suspension loaded with the probe without further treatment was used as the negative control. To be treated by the modified NWs, the bacterial suspension was flowed into the flow-through cell at a flux of 2 m$^3$ h$^{-1}$ m$^{-2}$. The bacterial suspension was also treated with 100 mg L$^{-1}$ Rosup reagent for 20 min to induce ROS generation as a positive control[62]. Aliquots of 100 μL were taken from each sample and were added into a 96-well black plate to determine the relative fluorescence intensity on a microplate reader (Molecular Devices, SpectraMax i3) with the excitation/emission wavelengths of 488/525 nm.

## Measurement of Cu concentration in the effluent

The concentration of Cu released in the effluent was measured by inductively coupled plasma mass spectrometry (ICP-MS, Thermo Fisher X Series). Briefly, a set of 15-mL aliquots were collected from the effluent at different sampling times. Before measurement, each aliquot was dosed with nitric acid to a concentration of about 1 wt% and filtered through a 0.45 μm membrane (Millipore, USA).

## Bactericidal performance of Cu$^{2+}$

A series of solutions containing 0.2, 0.4, 0.6, 0.8, 1.0 mg/L Cu$^{2+}$ were prepared using CuSO$_4$·5H$_2$O. Then the *E. coli* suspension ($10^9$–$10^{10}$ CFU mL$^{-1}$) was dosed into the Cu$^{2+}$ solutions to obtain a concentration of $10^6$–$10^7$ CFU mL$^{-1}$. These solutions were incubated for one hour and the live bacterial concentrations were measured using the plate count method. The bactericidal tests of different Cu$^{2+}$ concentrations were repeated for three times.

## Bactericidal test of the modified NWs at different flow rates

For the bactericidal test at the static condition (i.e., flow rate of zero), two pieces of copper foam with the modified NWs were first washed with pure water in the flow-through cell for two hours to remove surface impurities. Then the modified NWs were taken out and immersed in a 160-mL *E. coli* suspension with a concentration of $10^6$–$10^7$ CFU mL$^{-1}$, which corresponds to the treated volume in the flow-through cell at a flux of 2 m$^3$ h$^{-1}$ m$^{-2}$ for an hour. The live bacterial concentration was measured using the standard plate count method after being incubated for one hour. The bactericidal test at the static condition was repeated for three times. The bactericidal test in flow condition was conducted as introduced above. Specifically, the flow rate was controlled at 0.65, 2.7, and 7.8 mL min$^{-1}$, corresponding to a flux of 0.5, 2 and 6 m$^3$ h$^{-1}$ m$^{-2}$ respectively.

## Bacterial storage experiment

*E. coli* (CGMCC 1.3373), *P. aeruginosa* (CGMCC 1.12483), and *S. aureus* (CGMCC 1.12409) were cultured at 37 °C with shaking at 150 rpm for 12 h, washed to remove the nutrient medium and diluted with DI water to obtain a bacterial suspension of $10^6$–$10^7$ CFU mL$^{-1}$. The bacterial solution was flowed through the disinfection cell with the modified NWs at a flux of 2 m$^3$ h$^{-1}$ m$^{-2}$. The influent and effluent solutions were collected and stored at 25 °C under visible light illumination, which represented a typical natural aquatic environment. The bacterial concentrations of each samples were measured at a series of storage times from 0 h to 24 h.

## Disinfection in real water samples

The reclaimed water sample was collected from secondary effluent in a wastewater treatment plant (Shenzhen, China). The tap water was collected from the tap water faucet in the lab. The characteristics of the water samples are shown in Supplementary Table 5. Both were filtered through a 0.22 μm membrane to remove the indigenous bacteria. The *E. coli* stock solution was diluted in the two water samples to obtain a

concentration of $10^6$–$10^7$ CFU mL$^{-1}$. The prepared solutions were flowed through the disinfection cell with the modified NWs at a flux of 2 m$^3$ h$^{-1}$ m$^{-2}$. Storage experiments were conducted to evaluate the disinfection performance in real water samples.

## Long-term disinfection

To perform the long-term disinfection test, twelve chamber units were stacked in series (Fig. 4d inset), and each unit has an outer diameter of 90 mm and a thickness of 10 mm. Each chamber unit contained two pieces of copper foam with a size of 2 × 2 cm and a thickness of 2 mm. The effective filtration area for each chamber was 2.3 cm$^2$ and the effective filtration volume for twelve chambers was calculated to be 10.9 mL ($V_0$). *E. coli* stock solution was diluted with sterilized water to prepare the feed water with a concentration of $10^3$–$10^4$ CFU mL$^{-1}$. During the test, the feed water continuously flowed into the system and the flow rate was fixed at 2.7 mL min$^{-1}$. The feed water was replenished with a newly-prepared *E. coli* stock solution every two days, and the concentrations of live bacteria in the influent and effluent were measured by the standard plate count method. The time to spread the bacterial solutions on the plate was normalized to one hour after sampling. The treating capacity is defined as the ratio of actual treated volume ($V_{water}$) to the effective volume of the chambers ($V_0$). During the 30-day test, 116.64 L of the feed water was treated, which is over 10000 times that of $V_0$. Therefore, for a chamber volume of 1 L, an estimated 10,000 L feed water can be purified, which can provide the drinking water consumption of an adult (2 L per day) for more than ten years.

## Disinfection experiments with three different nanotip surfaces

ZnO nanorods were grown on the copper foam by a chemical precipitation method[63]. The cleaned copper foam (2 × 5 cm$^2$) was transferred into a Teflon-lined autoclave with 40 mL of an aqueous solution containing 4.5 g zinc nitrate hexahydrate, 2.1 g hexamethylenetetramine, and 2 mL ammonia, and heated at 90 °C for 15 h. The prepared sample was washed with DI water and dried in a vacuum oven.

Co, Mn-layered double hydroxides (LDH) nanoneedles were fabricated on nickel foam using a hydrothermal approach[64]. 0.72 g urea, 0.37 g NH$_4$F, 0.40 g MnCl$_2$·4H$_2$O and 1.2 g Co(NO$_3$)$_2$·6H$_2$O were dissolved in 80 mL DI water and transferred into a 100 mL Teflon-lined autoclave. A piece of nickel foam (2 × 5 cm$^2$) was cleaned and soaked in this solution. The autoclave was then sealed and placed in an oven at 120 °C for 6 h. Finally, the nickel foam was washed and dried in a vacuum oven.

Titanate nanowires were synthesized on a titanium (Ti) substrate by a one-step hydrothermal method[65]. A piece of Ti foam was first cleaned and transferred into a Teflon-lined autoclave with 40 mL of 1 M NaOH solution. Afterwards, the vessel was sealed and placed in an oven at 220 °C for 4 h. Finally, the prepared sample was washed and dried in a vacuum oven.

To prepare the modified ZnO nanorods, Co, Mn-LDH nanoneedles, and titanate nanowires, the same carbon-coating method was used as for treatment of the Cu(OH)$_2$ NWs. In the disinfection experiment, a bacterial suspension with $10^6$–$10^7$ CFU mL$^{-1}$ *E. coli* was flowed through the disinfection cell with the modified or unmodified nanostructured surfaces at a flux of 2 m$^3$ h$^{-1}$ m$^{-2}$. The live bacterial concentration in the effluent was measured at a normalized storage time of 5 h.

## All-atom molecular dynamics (MD) simulation

The amorphous carbon materials were constructed as follows. First, carbon atoms were randomly and approximately uniformly placed in a box of 6 nm × 6 nm × 6 nm to give a density of 3 g cm$^{-3}$. An amorphous carbon cube of 5 nm side was extracted from this structure. Then, parameters of sp$^2$ carbon atoms based on the INTERFACE force field were assigned to the amorphous carbon atoms[66], with a van der

Waals (vdW) diameter $\sigma_0 = 0.355$ nm and a well depth $\varepsilon_0 = 0.293$ kJ mol$^{-1}$. Finally, elastic network restraints were added to the bonds between adjacent carbon atoms to achieve a relatively rigid amorphous carbon material. CHARMM36m all-atom force field parameters[67] for 1-palmitoyl-2-oleoyl-sn-glycero-3-phosphoethanolamine (POPE) lipids, water model of transferable intermolecular potential 3 P and ions (Na$^+$, Cl$^-$) were used. CHARMM-GUI webserver[68] and GROMACS tools were used to set up all simulation systems[69], which were performed using GROMACS software (version 2019.4). For the interactions of materials with different vdW forces, we artificially adjusted the well depth ($\varepsilon$) of the carbon cube material ($1/4\varepsilon_0$, $1/2\varepsilon_0$, $7/8\varepsilon_0$, $15/16\varepsilon_0$, $2\varepsilon_0$). The dimensions of the initial box were 10.0 nm × 10.0 nm × 14.3 nm, which consisted of 1 amorphous carbon cubic, 338 POPE lipids, 30667 water molecules and 150 mM NaCl. The carbon cube was initially placed close to the POPE lipid bilayer and then went through a 100-ns free MD simulation. A time step of 2 fs, temperature of 310 K and periodic boundary conditions were used to all MD simulations. System snapshots and movies were generated by visual molecular dynamics[70].

## Simulation of bacterial motion inside a copper foam

Bacterial motion inside a porous copper foam was simulated by the software package ANSYS Fluent (2020 R2). First, we built a 3D model of the copper foam, and imported it into the ANSYS Fluent to solve the continuity. The domain of the computing model was set to be a cuboid with a size of 520 × 420 × 420 µm$^3$. The laminar flow inside the model was calculated by the Fluent computational fluid dynamics (CFD) code following the conservation of momentum and mass[71] given by:

$$\rho \frac{\partial \mathbf{u}}{\partial t} + \rho(\mathbf{u} \cdot \nabla)\mathbf{u} = \nabla \cdot [-p\mathbf{I} + K] + \mathbf{F} \quad (1)$$

$$\rho \nabla \cdot \mathbf{u} = \mathbf{0} \quad (2)$$

where $\rho$ is the density of the fluid, $\mathbf{u}$ is the velocity vector of the fluid, $t$ is the time, $p$ is the pressure of the fluid, $\mathbf{I}$ is the identity tensor, and $\mathbf{F}$ is the volume force vector. The viscous stress tensor $\mathbf{K}$ is defined by:

$$K = \mu(\nabla \mathbf{u} + (\nabla \mathbf{u})^T) \quad (3)$$

where $\mu$ is the dynamic viscosity, $T$ is the absolute temperature. The bacterial motion in the defied flow field was obtained by the discrete element method solver. For simplicity, the bacterium was approximated as a spherical particle, with a diameter of 1 µm. The motion of a bacterium in fluid flow is described by Newton's Second Law:

$$\frac{d}{dt}(m_p \mathbf{v}) = \mathbf{F}_D + \mathbf{F}_G + \mathbf{F}_b \quad (4)$$

where $m_p$ is the mass of the bacterium, $\mathbf{v}$ is the velocity of the bacterium, and $\mathbf{F}_D$, $\mathbf{F}_G$, $\mathbf{F}_b$ are the drag, gravity, and Brownian force, respectively. For small particles dispersed in a liquid, gravitational effect is negligible. The drag force is defined as:

$$\mathbf{F}_D = \left(\frac{1}{\tau_p}\right) m_p(\mathbf{u} - \mathbf{v}) \quad (5)$$

where $\tau_p$ is the bacterial velocity response time. The Brownian force that causes bacterial diffusion is given by:

$$\mathbf{F}_b = \zeta \sqrt{\frac{12\pi k_B \mu T r_p}{\Delta t}} \quad (6)$$

where $\Delta t$ is the time interval taken by the solver, $r_p$ is the particle radium, $k_B = 1.380649 \times 10^{-23}$ J K$^{-1}$ is the Boltzmann constant, and $\zeta$

(dimensionless) is a normally distributed random number with a mean of zero and unit standard derivation.

The left side of the model was defined as the velocity inlet and the particle generation surface, while the right side was defined as the outlet boundary. Symmetrical boundary conditions were assumed for the lateral sides of the domain. Bacteria were stuck to the wall of the foam model once they struck it. The influent velocity was $1\,mm\,s^{-1}$ and the flow was allowed to evolve until it was fully developed in the entire domain. 1681 bacteria were then released and allowed to move. The number of bacteria that collided with the walls of the copper foam and the number of bacteria that escaped from the outlet boundary were calculated.

### Types of contact between bacteria and nanotips during flow

The movement of the *E. coli* cells near the nanowires was simulated by a Brownian dynamics and computational fluid dynamics method[45], in which the information of the flow field near the nanotips was calculated by the CFD method and imported into the Brownian dynamics equation by which the movement of a bacterium was simulated.

The velocity of the flow field near the nanowires ($V$) was obtained by solving the Navier-Stokes equation:

$$\rho \frac{dV}{dt} = -\nabla p + \rho \mathbf{F} + \mu \Delta V \qquad (7)$$

where $\rho$ is the density, $p$ is the pressure, and $\mu$ is the viscosity of the fluid. A 5 × 5 tip array was used to present the nanowires on the copper foam. The cylindrical tips were perpendicular to the surface and the gap between two tips was set as $1\,\mu m$. The diameter of a tip was set as 200 nm and the length as $5\,\mu m$. We assumed the fluid impacted the nanotips at an angle of 45° to simplify the real conditions, and the velocity of the main flow was set as $5.5 \times 10^{-4}\,m\,s^{-1}$ corresponding to a flow rate of $2.7\,mL\,min^{-1}$ in the experimental conditions. No-slip boundary condition was set at the walls of the tips. The calculated velocity field is shown in Fig. 3c.

The *E. coli* cell was simplified using a bead-stick model with two balls to reflect both the translation and rotation (Supplementary Fig. 25). The velocity and location of the *i*th bead of the bead-stick model at a certain time was calculated by the Langevin equation:

$$\frac{dr_i}{dt} = V(r_i) + \frac{1}{\xi}\left[F_i^B(t) + F_i^S(t) + F_i^{EV,Wall}(t)\right] \qquad (8)$$

where $r_i$ is the position of the *i*th bead of bacterium, $V(r_i)$ is the velocity of the flow field at $r_i$, and $\xi$ is the drag coefficient of a bead. $F_i^B(t)$, $F_i^S(t)$, $F_i^{EV,Wall}(t)$ are Brownian force, spring force of the stick between two beads, and the repulsive force between the bead and walls, respectively. The simulation was terminated when the bacterium moved out the area of the flow field or collided with the tips. The movement of 100 bacteria was simulated. The number of bacteria that collided with the tip array was counted and the types of contact were determined.

### Stress distribution profile of bacterial cell envelope

The finite element method was used to explore the bacterial deformation, which was conducted using the software ABAQUS 6.11. The *E. coli* cell envelope was modeled as a spherocylindrical shell with a diameter of 500 nm and a length of $1\,\mu m$. The Young's modulus ($E$) encompasses the entire envelope with a value of 0.5 MPa obtained from AFM measurements (Supplementary Table 3).

The critical stress of the bacterial cell envelope was simulated according to the puncture test (Fig. 3a and Supplementary Fig. 17). The half opening angle of the MLCT-bio tip was set as 30° and the tip radius was 40 nm. The displacement of the tip was set as zero and the displacement of the cell was set as 100 nm. The stress profile of the cell envelope was then calculated and the maximum stress was found to be the critical stress required for rupturing the cell.

The stress distribution profiles of a bacterium under collision and tearing process were calculated. The nanowire was simplified as a cylindrical tip with a diameter of 200 nm. A single shell model was used for the *E. coli* cell as in the computation of the critical stress. Two types of contact (end-contact and middle-contact) between a bacterium and a tip were simulated and the displacement of the tip was set as zero. In the case of collision, when the work done by the tip equaled the complete kinetic energy loss of a bacterium, the cell encountered the maximum indentation from the collision process. The kinetic energy of a bacterium ($E_k$) is originated from the flow, which is determined by $E_k = 1/2mv^2$, where $m$ is the mass of a bacterium, and $v$ is its velocity. The maximum kinetic energy loss was calculated to be $2 \times 10^{-25}\,J$ and the maximum work done by the tip to a bacterium was obtained. The stress distribution profile under this case was calculated and compared with the critical stress for failure. In the case of tearing, the drag force of the flow was calculated, which equaled the adhesive force of the tip in the attached state. In a laminar flow, the drag force ($F_D$) can be determined by the classic Stokes law $F_D = 3\pi\mu d\Delta u$, where $\mu$ is the dynamic viscosity of the fluid, $d$ is the diameter of the particle and $\Delta u$ is the relative velocity of the fluid with respect to the particle[72]. Here, the diameter of the bacterium was assumed to be $0.5-1\,\mu m$, and $F_D$ was calculated to be $2.3-4.7 \times 10^{-13}\,N$. The stress distribution profiles were simulated under this reaction force in the two types of contact and compared with the critical stress for failure.

### Reporting summary

Further information on research design is available in the Nature Portfolio Reporting Summary linked to this article.

## Data availability

All data are available in the main text and the supplementary information. Source data are provided with this paper.

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

## Acknowledgements

This study was supported by National Natural Science Foundation of China (No. 52022049, Q.Y.W. and No. 52221004, H.Y.H.), the Shenzhen Science, Technology and Innovation Commission (No. RCJC20221008092758099, Q.Y.W.) and Tsinghua Shenzhen International Graduate School (JC2022014, Q.Y.W. and C.Y.). We thank H.M. Cheng at Institute of Metal Research, Chinese Academy of Sciences for valuable discussion and P. A. Thrower for reviewing the manuscript. We also acknowledge W.Q. Wang at the State Key Laboratory of Tribology, Tsinghua University, for technical support in the atomic force microscopy tests, and S.Y. Wu at Fujian Agriculture and Forestry University for suggestions on finite element method simulations.

## Author contributions

Conceptualization: C.Y., L.P., H.Z.; Methodology: L.P., H.Z. H.H., C.Y., Q.W.; Experiment: L.P., H.Z., H.W., Z.G.; Visualization: H.Z., H.W., Z.G.; Funding acquisition: Q.W., C.Y.; Supervision: H.H., C.Y., Q.W.; Writing – original draft: L.P., H.Z., C.Y.; Writing – review & editing: L.P., H.Z., H.H., C.Y., Q.W.

## Competing interests

The authors declare no competing interests.
