## [Peer Review File · Nature Communications]

Hydrodynamic tearing of bacteria on nanotips for sustainable water disinfectionEditorial Note: Parts of this Peer Review File have been redacted as indicated to maintain the confidentiality of unpublished data.

Reviewer #1 (Remarks to the Author):

The manuscript by Peng et al. describes a hydrodynamic bactericidal mechanism that is remarkably effective in the inactivation of bacteria in the water. The results indicate that the bacterial cell envelope was torn when interacting with high aspect ratio surface nanostructures. Such interactions are shown to be a consequence of the increased London dispersion forces between the bacteria and the surface nanostructures. There are, however, a number of questions and issues that need to be addressed to improve the present study.

1) The adhesion force between the carbon-coated AFM tip and the bacterium has been reported to be in the range of 0.5-2 nN. I believe it is crucial to report it with more statistical details (e.g., mean +/- SD based on repeated measurements) to specify a more accurate adhesion force required for the adhesion of bacteria to nanostructures.

2) Regarding the bactericidal mechanism and considering the extensive literature on mechanobactericidal nanostructures, the role of nanostructure geometry in your system has to be further studied and discussed. For instance, you have stated that London dispersion interaction is the critical factor regardless of the type of surface with nanotip characteristics. Yet you have shown disparities in the removal ability when the geometry and chemical composition of the nanostructures varied (Fig. 4e). If the carbon coating is applied to all different types of nanostructures, it can be hypothesized that geometry still plays a significant role in the bactericidal properties of the surface.

3) Continuing on the role of geometry, I think it is necessary to systematically vary the dimensions and aspect ratio of the nanostructures to confirm your hypothesis that London dispersion interaction is the critical factor in the bactericidal mechanism. If the two types of nanostructures with significantly different aspect ratios (e.g, one with an aspect ratio of 2 and one with an aspect ratio of 20) have a similar London dispersion interaction with the bacteria, would the bactericidal efficiency be similar as well?

4) In addition to the previous comment, I was wondering if the authors observed/considered the bending of the nanostructures when interacting with bacteria. Storage and release of mechanical energy in high aspect ratio nanostructures have been previously shown to damage bacteria (Linklater et al., ACS Nano, 2018). The stiffness and height of the nanostructures play an important role in this regard. Investigating a similar mechanism is crucial to identify the determining factors in the bactericidal activity of the present system.

5) A remarkable bactericidal efficiency against Gram-positive *S. aureus* has been reported but no SEM/TEM images of the bacteria in the influent and effluent water have been presented. As *S. aureus* has a very different geometry and structure compared to *E. coli*, it would be very interesting to see whether it undergoes similar damage. Specifically, how does *S. aureus* collide by end-contacts and what would be the stress distribution profile in a simulation? More importantly, is the adhesion force of *S. aureus* to the nanostructures different from that of *E. coli*? I am curious to know how much force is required to tear the thicker cell wall of *S. aureus* in an aqueous medium.

6) The authors did not consider oxidative stress as a factor contributing to the bactericidal properties of their system. It has been shown that when bacteria are exposed to extreme topographies, they start producing reactive oxygen species to an extent beyond their tolerance (Jenkins et al., Nature Communications, 2020). I advise the authors to investigate this effect in their work with proper control conditions to delineate the role of oxidative stress (in case it exists) from tearing caused by London dispersion interactions.

7) In preparing bacterial samples for SEM, critical point drying is usually used as the last step of dehydration in similar studies. However, the authors have used freeze-dried bacteria before SEM. I am concerned that freeze-drying would have an effect on the morphology of the bacteria. Further evidence is needed to confirm that the morphology is not altered at that stage.

8) As changes in the flow rate affect the hydrodynamic boundary layer close to the surface, testing the developed water disinfecting system at lower and higher flow rates is essential to show its

potential for different applications.

9) On page 5, lines 173-176, it has been stated that SYTO 9 stains intact live cells while propidium iodide stains bacteria with damaged membranes. Please note that SYTO 9 stains both live and dead bacteria as it is a membrane-permeable dye. Therefore, the abovementioned statement has to be edited in the manuscript.

10) The Discussion is limited to repeating the conclusions of the study and does not address its possible limitations/drawbacks with regard to the literature.

11) There are inconsistencies in reporting the suppliers in the Methods section. Please provide the name of all suppliers and the catalog numbers of materials and reagents used in the study.

Reviewer #2 (Remarks to the Author):

The manuscript reports a hydrodynamic-bactericidal approach in the application of water disinfection. The authors propose a unique bactericidal mechanism from the mechanics' perspective using both simulation and experiments and concluded that 'tearing effect was the leading cause of bacterial destruction'. The work is of interest to the research communities that aim at developing antimicrobial-free disinfection technology to address global challenges in environment and healthcare. However, there are a few fundamental questions that need to be answered or clarified before it is accepted for publication.

1. The POPE lipid bilayer model is over-simplified for the modelling of bacterial cell wall as the cell wall structure of prokaryotic cells is very different from that of eukaryotic cells. A polysaccharide (peptidoglycan) layer is present in the bacterial cell wall which is a rigid and load-bearing component. This layer is vital to the viability of bacteria and more pronounced for the gram-positive bacteria. The simulation of bacterial cell wall mechanics should take this into consideration.

2. The physiochemical properties of carbon coated nanowires should be fully characterised, especially their chemistry, surface charge, wetting and their contribution to the London dispersion force. It is unclear how the amorphous carbon layer on the nanowire surface changed the magnitude of the London dispersion force and why it had a strong attraction to the bacteria, leading to the increased bacterial adhesion force.

3. It is necessary to check whether the Cu(OH)₂ nanowires were unchanged or converted to something else e.g. CuO or Cu₂O after the high-temperature pyrolysis of coated glucose at 550°C for 2h in Ar to form carbon coating. It is important to have a detailed information of both as-synthesised Cu(OH)₂ nanowires and glucose after heat treatment to fully understand the physiochemical characteristics of coated nanowires because they are crucial in elucidating adhesion and bactericidal mechanisms.

4. Given widely reported studies of the role of reactive oxygen species (ROS) induced by carbon-based nanomaterials (nanotubes, graphene nanosheets, quantum dots), it is suggested to measure the ROS level on the C-NW surfaces, so that other bactericidal mechanisms than mechano-killing could be ruled out.

5. Explain why the water disinfection experiments were conducted under visible light illumination. What would be the implication if the carbon coated NWs were photocatalytic? CuO and Cu₂O are known photocatalysts that can generate ROS. If combined with carbon, they may even form heterojunctions which are likely to have excellent photocatalytic properties under visible light.

6. Did Fig.4e also imply other possible killing mechanisms apart from the mechano-bactericidal mechanism? If not, how the nano-feature size affects the bacterial killing?

7. Explain how gram-positive bacteria with spherical shape were teared. Any evidence to support it? As the NWs on 3D porous Cu substrates are not aligned to a specific direction, from Fig.2n, piercing rather than tearing seemed to be a more likely mechanism.

Reviewer #3 (Remarks to the Author):

Peng et al. report a copper-based nanostructured material that is highly effective at inactivating bacteria through a hydrodynamic tearing mechanism. The authors use modeling and AFM measurements to design nanostructures that incorporate a London dispersion interaction. They construct carbon-based nanostructures within a porous copper foam and show that it removes bacteria from liquid culture by over 6 log-fold, with no trace of living bacteria detected. Using SEM and TEM, they find that bacteria have a significant amount of their membranes removed and that a significant amount bacterial debris is left on the nanostructures. The authors show significant propidium iodide fluorescence within the bacterial cytoplasm following treatment. By estimating flow velocities, cell wall rigidity, and interaction forces with the nanostructures, the authors propose that bacteria are torn apart by nanostructures rather than disrupted mechanically by collision. Finally, they show that the mechanism works against other bacteria including *P. aeruginosa* and *S. aureus*, and that nanostructures using other materials including ZnO, Co, and Titanate are effective at removing bacteria.

Overall, I find the potential result of high-performing microbial removal to be a very exciting result that could advance the field of antimicrobial nanostructure surfaces in a significant way. The authors should also be lauded for their efforts that use different mixed technical approaches including AFM, biophysical modeling, and materials synthesis. However, I have major concerns about the microbiological characterization, lack of details throughout, and the inconsistency of the antimicrobial activity data. In addition, the data suggest contradictory rupturing mechanisms and the central conclusion of the manuscript rests on the assumption that bacteria are killed by the nanostructures rather than being adsorbed by them. Sufficient quantitative data that rules out adsorption is not provided.

Major concerns:

1. Fundamental details of the bacterial growth and killing assays are lacking, which make the experiments difficult to interpret and impossible to reproduce. Examples include but are not limited to a lack of description of the growth medium that was used to culture the bacteria, the media that was used to count CFUs, how long bacteria remained resuspended in DI water following growth, what volume of culture was treated, and over what time interval the treatment was performed. A cell density of 10^9 - 10^{10} CFU/mL is listed as a description of exponential phase. However, this is not a qualifier of exponential growth.
2. Mechanism of bacterial rupturing. The fluorescence data and SEM/TEM data are in seeming contradiction with each other. The authors observe significant propidium iodide using epifluorescence and structured illumination (Fig. 2h and Supp. Fig. 9), which support their claim that the bacterial membrane is ruptured. Propidium iodide stains nucleic acids in the cytoplasm. However, the SEM and TEM show that cells are ruptured with defects on the order of 100-200 nm (Fig. 2j and 2l), for which the authors indicate the cells have "cytoplasm missing" (line 187). The two interpretations are contradictory because the cytoplasm cannot both be missing due to large ruptures and be stained with propidium iodide. Given that cells are resuspended in pure water, pores of this size would result in the loss of nucleic acids and cytoplasmic content. An analysis that determines the fraction of cells that are propidium iodide positive or have loss of cytoplasm is necessary to further support the claim. Data quantifying the distribution of rupture sizes would also be helpful but are given.
3. The effect of flow on bacterial tearing seems contradictory. The model presented in Fig. 3 shows most of the rupturing force is due to the London dispersion interaction and that flow-induced collision produces a negligible amount of pressure on the bacterial membrane ($2.6E-4$ MPa / $699E-4$ MPa or 0.3% in the tip interaction in Fig. 3e vs. 3g). This would suggest that flow is not necessary for bacterial ruptures. However, the authors show that the modified nanostructures are largely ineffective without flow (Suppl. Fig. 12), raising an apparent contradiction.
4. Bacteria are cultured to a density of 10^9 - 10^{10} CFU/mL, which suggests growth in a rich medium. Subsequent resuspension into pure water could sensitize bacteria a number of non-physiological ways including nutrient and osmolarity shocks. Is the bacterial removal effect by nanostructures observed if bacteria are cultured in a low osmolarity / low nutrient environment or introduced into the device using the same growth medium instead of pure water?
5. The central conclusion of the paper rests on the assumption that cells are killed by the nanostructures. An alternative explanation for the results is that bacteria are largely adsorbed to

the nanostructures but this scenario is not adequately addressed. The authors state that 360-400 cells per image was observed in the influent and effluent (lines 177-199). However, this spot analysis is not sufficient to support the conclusion, as the cell density in microscopy images is highly variable and does not provide adequate statistical sampling. The authors should provide an appropriate sampling analysis to quantify the number of cells in the influent and effluent in all treatments, which should range in the 10^6 - 10^7 CFU/ml.

6. The removal efficacy is listed but no units are given (Fig. 2f). Are these CFUs / mL? Absolute CFUs? If so, what volumes are treated in each condition? This is data that is critical to the conclusion but insufficient information is provided to properly evaluate the effect. The full CFU data for both the influent and effluent should be provided.

Other issues:

1. The bacteria in Fig. 2k does not appear to be exponential phase *E. coli* and the cells do not have the same shape or aspect ratio as the cells in the SEM image (Fig. 2i). This raises a concern that the microbial growth conditions may not be consistent between experiments.

2. It is unclear to what extent pores are present in Supp. Fig. 9 and how representative the structured illumination image is. The sample size is not large enough to give statistical significance and it is unclear whether the sections of the DIO fluorescent image indicate a pore is present, are due to irregular staining, or due to adjustment of the lookup tables. In addition, one of the treated cells is not rod-shaped, which does not match the EM images in 2j and 2l or the claim that the cell is from exponential phase.

3. The resolution of the epifluorescence images in Fig. 2h is insufficient to make a conclusion. The data should be quantified to determine which fraction of cells in the overall effluent is positive for propidium iodide.

4. Given a rate of 2.7 mL/min, it would take approximately 12 hours to filter the ~2L bacterial resuspension in their setup (Supp. Fig. 8). During this time, *E. coli* in pure water may cease to swim due to the lack of nutrients. However, it does not appear that the influent is homogenized by any type of stirring mechanism. Are all experiments (modified NW, modified Cu, Cu(OH)₂ NW) performed over the same time period with homogenization, using the same volume, and same rate of flow for all devices? Information about these conditions are not given.

5. Many figure captions contain insufficient detail. For example, it is unclear from the caption alone what the velocity field boundaries are in Fig. 3C, what the numbers in red in Fig. 3e-h represent, and what the 0.05 on the nanotip represents.

6. A bacterial culture density of 10^6 - 10^7 CFU/mL is used in the device. Is this culture density critical? Are the nanostructures effective at higher or lower culture densities? Does the undiluted 10^9 - 10^{10} CFU/mL culture overwhelm the device?

7. If the London dispersion is responsible for rupturing, does this suggest that ions in solution would negate the effect? It would be helpful if the authors addressed this issue to understand potential limitations of the device.

Reviewer #4 (Remarks to the Author):

The elimination of pathogens from water has long been a major challenge. In this study, the authors investigated the hydrodynamic tearing of bacteria on nanotips for sustainable water disinfection. A rapid inactivation of >99.9999% of the bacteria in water in a 30-day field test was observed. The authors proposed that the mild water flow can efficiently tear up the bacteria through the higher dispersion force between the nanotip surface and the cell envelope. It was further demonstrated that tearing of the cell-envelope led to the rupture of the bacteria rather than collisions.

Since the effective mechanical disinfection in bulk water has not been investigated, this manuscript provided an innovative approach for water disinfection. This manuscript provides a valuable contribution to the topic and thus merits publication after minor revisions. A few comments/concerns are shown below for the authors' consideration.

1. The disinfection effectiveness was evaluated at a flux of $2 \text{ m}^3 \text{ h}^{-1} \text{ m}^{-2}$. How will the flow rate of

water effect the disinfection efficacy?

2. As the authors have mentioned the energy consumption of conventional disinfection practices (chemicals, UV, etc). Can the authors provide some insights on the comparison of hydrodynamic tearing of bacteria on nanotips with conventional disinfection practices in terms of energy consumption. This is extremely important with regard to its practical application.

3. For the long-term disinfection experiment, diluted *E. coli* stock solution was applied as the feed water. However, colloids, particles, dissolved organic matter, and ions coexist with pathogens in realistic water treatment conditions. Will these co-existing substances influence the interaction between pathogens and the modified NWS?

4. Did the author investigate the potential leakage of nanotips during operation?

5. Line 209-219. As the disinfection mechanisms are different among metal nanoparticles, nanostructured surfaces, and the modified NWS, is the different contact time required for effective disinfection?

6. Fig. 2 (o). For carbon nanomaterials, they required long contact time while the log removal values were relatively low. Which factor is responsible for the different performances of carbon nanomaterials and modified NWS with carbon coating in this study?

7. Line 320-321, there was bacterial debris accumulated in the first unit during the 30-day operation. Will this cause the biofouling of nanotips? Will the co-existing substances (colloids, etc) in water accelerate the accumulation of cell debris?

8. Line 323-325, limitation might exist here, as this calculation was based on the disinfection performance with diluted *E. coli* solution as the feed water.

9. Line 335-339, substantially different nanostructures were observed with $\text{Cu}(\text{OH})_2$, ZnO , LDH, and titanate as support layers, which would certainly influence the interaction between cell and the surfaces. More discussions may be needed.

10. Line 352-357, inactivation of virus with a much smaller size was not investigated in this study. It would also have been better not to mention "exploiting fluidic energy to destroy pathogens" since only tearing of bacteria was studied. To increase the broad impact of this work, the limitation for the virus inactivation may be mentioned.

Response to Reviewers

We appreciate the detailed and constructive comments provided by the reviewers. The reviewer comments are laid out below in italic and blue, and our response is given in normal font.

Reviewer #1:

The manuscript by Peng et al. describes a hydrodynamic bactericidal mechanism that is remarkably effective in the inactivation of bacteria in the water. The results indicate that the bacterial cell envelope was torn when interacting with high aspect ratio surface nanostructures. Such interactions are shown to be a consequence of the increased London dispersion forces between the bacteria and the surface nanostructures. There are, however, a number of questions and issues that need to be addressed to improve the present study.

Response: We appreciate the reviewer's positive comments. The questions and issues raised by the reviewer are addressed point by point below.

1) The adhesion force between the carbon-coated AFM tip and the bacterium has been reported to be in the range of 0.5-2 nN. I believe it is crucial to report it with more statistical details (e.g., mean +/- SD based on repeated measurements) to specify a more accurate adhesion force required for the adhesion of bacteria to nanostructures.

Response: We appreciate the reviewer's valuable suggestion. The adhesion force has been specified as 0.9 ± 0.5 nN (mean \pm SD from seven different cells) according to Supplementary Fig. 6.

2) Regarding the bactericidal mechanism and considering the extensive literature on mechanobactericidal nanostructures, the role of nanostructure geometry in your system has to be further studied and discussed. For instance, you have stated that London dispersion interaction is the critical factor regardless of the type of surface with nanotip characteristics. Yet you have shown disparities in the removal ability when the geometry and chemical composition of the nanostructures varied (Fig. 4e). If the carbon coating is applied to all different types of nanostructures, it can be hypothesized that geometry still plays a significant role in the bactericidal properties of the surface.

Response: Thank the reviewer for the insightful comment. Considering the impact of

nanostructure geometry, representative nanotip arrays were synthesized on porous substrates with significant geometric differences, including Cu(OH)₂ NWs, ZnO nanorods, Co, Mn-LDH nanoneedles and titanate nanowires, which are applicable for water treatment and can be reproduced easily. Before carbon coating, the four nanomaterials showed weak bactericidal performance during flow (~1 log), which suggests that the geometry and chemical composition of the nanostructures were not the key factors determining the rupture of bacteria in flowing water. By contrast, after carbon coating to elevate the London dispersion interaction, the bactericidal performance of the four nanomaterials was largely increased by at least three orders of magnitude (Fig. 4e). Thus, the important discovery in our study is that by introducing the nanotip surface with a suitable van der Waals well depth parameter (e.g. carbon coating), we transform the prevalent interaction mode of bacterium-surface collision into a strong cell-envelope tearing process, which is the key reason of the rapid inactivation of bacteria in water.

Fig. 4e | Bactericidal performance of the unmodified and modified ZnO nanorods, Co, Mn-LDH nanoneedles and titanate nanowires.

On the other hand, Fig. 4e also shows the differences in the bactericidal performance among the carbon-coated nanomaterials, indicating that after achieving a higher London dispersion interaction at the contact surface, the nanostructure geometry still has influence on the forces exerted to the bacteria. To give more information about the effects of geometry, we have counted the geometrical parameters of the four kinds of nanotip arrays (Supplementary Table 5). Potential influence of geometry in our system is discussed in the response for the following comments (comment 3 and comment 4), which has also been included in the revised

manuscript (page 10, line 380–389; page 11, line 397–404).

3) Continuing on the role of geometry, I think it is necessary to systematically vary the dimensions and aspect ratio of the nanostructures to confirm your hypothesis that London dispersion interaction is the critical factor in the bactericidal mechanism. If the two types of nanostructures with significantly different aspect ratios (e.g, one with an aspect ratio of 2 and one with an aspect ratio of 20) have a similar London dispersion interaction with the bacteria, would the bactericidal efficiency be similar as well?

Response: Our authors believe that this is a constructive and interesting comment. Yet currently, the fabrication of nanomaterials with precisely-controlled geometry is limited, especially on porous three-dimensional substrates. Considering the application in water treatment, we have synthesized four kinds of nanotip arrays on porous substrates, which exhibit relatively low toxicity in water and can be reproduced easily. To give more information about the influence of geometry, we summarized the length, diameter, and tip density of the four kinds of nanotip arrays used in this study in Supplementary Table 5, which show significant geometric differences.

Supplementary Table 6 | Geometrical parameters of the nanomaterials used in this study

Nanomaterial	Density (30 μm^2)	Length (nm)	Diameter (nm)	Aspect ratio
Cu(OH) ₂	165	5326 \pm 890	199 \pm 32	26.6
ZnO	135	2123 \pm 539	340 \pm 70	6.2
Co, Mn-LDH	195	2883 \pm 700	127 \pm 20	22.7
Titanate	175	2911 \pm 617	47 \pm 8	61.9

Geometrical data were collected from at least 30 individual nanotips in SEM images.

As can be seen from Fig. 4e, when the surface London dispersion interaction was elevated by carbon coating, the nanostructure geometry still had influence on the bactericidal property of the nanotips. Specifically, the carbon-coated ZnO nanorods with an aspect ratio of about 6 showed lower bactericidal efficiency than the carbon-coated Co, Mn-LDH nanoneedles and

titanate nanowires with higher aspect ratios (Supplementary Table 6). This is because ZnO nanorods had larger diameters and a lower tip density. Typically, nanostructures with a blunt feature are supposed to deliver less mechanical stress¹, due to an enlarged contact area. Besides, a lower tip density of ZnO nanorods also reduces the possibility of bacterial contact with the nanotips during flow, which negatively impacts their bactericidal performance.

In this work, we have demonstrated a unique hydrodynamic tearing mechanism of bacteria during flow. From this regard, the tip diameter and curvature that determine the adhesion force at cell-nanotip contact surface are important geometrical parameters governing the bactericidal properties of the surface. Exploring the influence of geometrical parameters in this new force model is a meaningful task. However, the fine adjustment of geometrical parameters of nanostructures is currently a challenging problem, especially on porous substrates. We believe the advancement in nanofabrication techniques such as reactive ion etching or deep UV lithography may help to provide more precise analysis on the role of nanostructure geometry, which will be carried out in the future.

4) In addition to the previous comment, I was wondering if the authors observed/considered the bending of the nanostructures when interacting with bacteria. Storage and release of mechanical energy in high aspect ratio nanostructures have been previously shown to damage bacteria (Linklater et al., ACS Nano, 2018). The stiffness and height of the nanostructures play an important role in this regard. Investigating a similar mechanism is crucial to identify the determining factors in the bactericidal activity of the present system.

Response: We understand the concern raised by the reviewer, yet after carefully reading the referred publication, we believe the bactericidal mechanism that Linklater *et al.* reported in 2018 is different from ours. The reasons are listed below.

First, the dimensions of the nanostructures that Linklater *et al.* used were different from ours, therefore, the physical rupture of bacteria cannot be explained by the same mechanical model. In the work by Linklater *et al.* (2018), they used *multiwalled carbon nanotubes (CNTs) with a super high aspect ratio (100–3000) between their length (microns) and diameter (approximately 10 nm)*. Owing to this geometry, the CNTs are flexible and can easily deform

*in contact with the bacterial cells*². By contrast, the nanotips in our system have much larger diameters, and the aspect ratios of the nanomaterials are far lower than the CNTs. For example, the Cu(OH)₂ NWs have a diameter around 200 nm and a length about 5 μm (Supplementary Table 6), causing them to be rigid and deflect little laterally when interacting with bacteria. Similarly, in Linklater's other studies, they have reported that the nanospikes with relatively small aspect ratios (5–30) were recognized as rigid nanostructures and were not perceived to deflect substantially^{3,4}.

Moreover, the stiffness and height of the nanostructures do not significantly change the bactericidal properties of different nanomaterials in flowing environment. As is shown in Fig. 4e, the nanomaterials with different aspect ratios showed weak bactericidal efficiencies (~ 1 log) before carbon coating. By contrast, after carbon coating to elevate the London dispersion interaction, the nanomaterials showed largely increased bactericidal performance. We demonstrated that only when the nanotip surface possesses a sufficient van der Waals well depth parameter to allow a strong London dispersion interaction can we enable a fast physical killing of bacteria during flow, which is the primary finding of our work.

5) A remarkable bactericidal efficiency against Gram-positive S. aureus has been reported but no SEM/TEM images of the bacteria in the influent and effluent water have been presented. As S. aureus has a very different geometry and structure compared to E. coli, it would be very interesting to see whether it undergoes similar damage. Specifically, how does S. aureus collide by end-contacts and what would be the stress distribution profile in a simulation? More importantly, is the adhesion force of S. aureus to the nanostructures different from that of E. coli? I am curious to know how much force is required to tear the thicker cell wall of S. aureus in an aqueous medium.

Response: We appreciate the reviewer's valuable questions. In response to the reviewer's interests, we have examined the morphological change of *S. aureus* before and after treatment using the SEM/TEM and observed similar mechanical damage as the *E. coli*. As seen from the SEM images, the initial *S. aureus* cells had intact cell membranes; however, after being treated by the modified NWs, the cell membranes were disrupted with holes on the surface

(Supplementary Fig. 20a,b). The TEM images also indicate that the cell envelope structure was damaged (Supplementary Fig. 20c,d). However, there was only a minor leakage of cytoplasm, suggesting that *S. aureus* is more resistant to mechanical damage compared with Gram-negative *E. coli*, due to a thicker peptidoglycan cell wall⁵.

Supplementary Fig. 20 | Morphological change of *S. aureus*. a,b SEM images of the initial *S. aureus* (a) and *S. aureus* treated by the modified NWs (b). c,d, TEM images of the initial *S. aureus* (c) and *S. aureus* treated by modified NWs (d). (Scale bar = 200 nm)

We further investigated the forces exerted on the *S. aureus* cell during the collision and tearing processes. Because *S. aureus* is a coccus, we assume the cell envelope of the *S. aureus* as a spherical shell. Therefore, end-contact does not exist in the case of *S. aureus*. [Redacted]. The stress distribution profiles of the *S. aureus* under a typical contact form were obtained by the finite element simulation [Redacted]. [Redacted].

Considering the different cell envelope composition between the Gram-negative *E. coli* and Gram-positive *S. aureus*, we further performed a molecular dynamics (MD) simulation to study whether there is strong adhesion between the *S. aureus* cell and the surface of the carbon-coated nanowires. [Redacted].

Redacted

[Redacted].

Based on the above results, we can conclude that the *S. aureus* cells can be damaged after flowing through the modified NWs, and that the rupture mechanism of the *S. aureus* is similar to the *E. coli*. However, we also observed the extent of rupture, the stress exerted on the cell envelope and the level of London dispersion interaction in the case of Gram-positive *S. aureus* were different from Gram-negative *E. coli*. We have made discussions about the difference between the *E. coli* and *S. aureus* in the revised manuscript, including cell shapes, cell mechanics and cell envelope composition (page 11, line 405–413). We believe the influence of bacterial species on hydrodynamic-bactericidal mechanism is an interesting question, and need to be thoroughly investigated in the future.

Redacted

[Redacted].

6) *The authors did not consider oxidative stress as a factor contributing to the bactericidal properties of their system. It has been shown that when bacteria are exposed to extreme*

topographies, they start producing reactive oxygen species to an extent beyond their tolerance (Jenkins et al., Nature Communications, 2020). I advise the authors to investigate this effect in their work with proper control conditions to delineate the role of oxidative stress (in case it exists) from tearing caused by London dispersion interactions.

Response: Thanks for the reviewer’s comment. We have measured the intracellular reactive oxygen species (ROS) in the bacteria treated by the modified NWs. The untreated *E. coli* sample was used as a negative control, while the Rosup reagent, provided by the supplier, was used to stimulate the bacteria to generate intracellular ROS as a positive control (see Supplementary Methods). As is shown in Supplementary Fig. 15, the ROS production was not elevated in the bacteria treated by the modified NWs, compared to the negative and positive control. Therefore, the contribution of oxidative stress in our system is negligible. Relevant discussion about ROS has been added in the revised manuscript (page 7, line 225–227).

Supplementary Fig. 15 | Measurement of the intracellular ROS in the untreated *E. coli* cells and the *E. coli* cells treated by the modified NWs. The *E. coli* sample treated by Rosup was used as a positive control.

The ROS measurement indicates that the bacterial inactivation process in our system is different from the previous work (Jenkins et al., 2020). In our system, the bacteria were torn up by an instantaneous contact with the carbon-coated nanotips during flow. After flowing out, the bacteria were severely ruptured and left with pores (Fig. 2j and 2l), leading to complete inactivation. By contrast, in the work by Jenkins et al (2020)⁵, the bacteria were incubated on the nanopillar surfaces. As they said, “*The nanopillars induce deformation and penetration of*

the bacterial cell envelope, but do not rupture or lyse bacteria". Following a 24-h incubation, the mechanical stress exerted by nanopillars induced increase of oxidative stress, which causes the time-dependent reductions in bacterial viability in their system. A comparison between our work and the previous study was listed in the table below (Table R1).

Table R1 | Comparison between our work and Jenkins *et al.*, (2020)

Aspect	This work	Jenkins et al. , (2020)
Experimental condition	Continuous flow	Static incubation
Contact time	7 s	10–24 h
Bactericidal effect	Complete inactivation	Reduce the capacity of bacteria to proliferate
Bacterial morphology after treatment (E. coli)		
7) In preparing bacterial samples for SEM, critical point drying is usually used as the last step of dehydration in similar studies. However, the authors have used freeze-dried bacteria before SEM. I am concerned that freeze-drying would have an effect on the morphology of the bacteria. Further evidence is needed to confirm that the morphology is not altered at that stage.

Response: The critical point drying and tert-butyl alcohol freeze-drying are both recommended drying processes for preparing microbial samples⁹. To address the reviewer's concern, the bacterial samples for SEM were also prepared by critical point drying. Specifically, after fixation with 2.5% glutaraldehyde and dehydration with a series of ethanol/water solutions, the bacterial samples were critically point dried using a Leica EM CPD300, which were then observed with the SEM after being sputter-coated with platinum.

Figure R3 shows the SEM images of the samples prepared by critical point drying. As can be seen, the initial *E. coli* cells were rod-shaped with intact cell membranes. By contrast, the *E.*

coli cells treated by the modified NWs were ruptured and had pores on the surface. This is consistent with the morphology of bacteria prepared by freeze-drying (Fig. 2i,j). Based on these results, we confirmed that the morphology of bacteria was not altered during tert-butyl alcohol freeze-drying.

Fig. R3 | SEM images of the bacterial samples prepared by critical point drying. a, The initial *E. coli* cells. b, The *E. coli* cells treated by the modified NWs (scale bar = 500 nm).

8) As changes in the flow rate affect the hydrodynamic boundary layer close to the surface, testing the developed water disinfecting system at lower and higher flow rates is essential to show its potential for different applications.

Response: According to the reviewer's suggestion, we have tested the disinfection performance of the modified NWs at different flow rates (Supplementary Fig. 19). The bactericidal test in the static condition showed limited bactericidal activity (~ 1.4 log), indicating that the significance of water flow to the rupture of the bacteria. In the flow condition, the modified NWs achieved remarkable inactivation of $>99.9999\%$ *E. coli* (>6 log removal) in water at a flux of $0.5\text{--}2\text{ m}^3\text{ m}^{-2}\text{ h}^{-1}$, which is the commonly-adopted flux in filtration modules for water treatment^{10,11}.

Supplementary Fig. 19 | Influence of different flow rates on the inactivation of bacteria by the modified NWs. The flow rate is given in the form of flux ($\text{m}^3 \text{h}^{-1} \text{m}^2$). Flow rate of zero represents the bactericidal test of the modified NWs in the static condition.

When the flux was further increased to $6 \text{ m}^3 \text{ m}^{-2} \text{ h}^{-1}$, we observed a decrease in the inactivation efficiency. This is because at such a high flux, the contact time of the bacteria within the materials was decreased, leading to a substantial decrease in the contact possibility between the bacteria and nanotips. Similar discussion about the influence of flow rate has been added in the revised manuscript (page 9, line 323–332).

9) On page 5, lines 173-176, it has been stated that SYTO 9 stains intact live cells while propidium iodide stains bacteria with damaged membranes. Please note that SYTO 9 stains both live and dead bacteria as it is a membrane-permeable dye. Therefore, the abovementioned statement has to be edited in the manuscript.

Response: We thank the reviewer’s comment. The statement has been revised as “*Bacteria with intact cell membranes were stained with SYTO 9 (green), whereas nonviable bacteria with damaged membranes were stained with propidium iodide (red).*”

Guidance from the supplier’ manual is provided here for your information: “*When used alone, the SYTO 9 stain generally labels all bacteria in a population—those with intact membranes and those with damaged membranes. In contrast, propidium iodide penetrates only bacteria with damaged membranes, causing a reduction in the SYTO 9 stain fluorescence when both dyes are present. Thus, with an appropriate mixture of the SYTO 9 and propidium iodide stains, bacteria with intact cell membranes stain fluorescent green, whereas bacteria with*

damaged membranes stain fluorescent red.

(<https://www.thermofisher.cn/document-connect/document-connect.html?url=https://assets.thermofisher.cn/TFS-Assets%2FMSG%2Fmanuals%2Fmp07007.pdf>)

10) The Discussion is limited to repeating the conclusions of the study and does not address its possible limitations/drawbacks with regard to the literature.

Response: According to the reviewer's suggestion, more discussions on potential limitations of the present study and implications for the future work are provided in the revised manuscript, including further exploration of the role of nanostructure geometry and the influence of bacterial species, potential limitation on the inactivation towards other pathogens and more studies under real-world conditions (page 11–12, line 397–434).

11) There are inconsistencies in reporting the suppliers in the Methods section. Please provide the name of all suppliers and the catalog numbers of materials and reagents used in the study.

Response: The suppliers and catalog numbers of the materials and reagents used in this study have been checked carefully and provided in the Methods section.

Reviewer #2:

The manuscript reports a hydrodynamic-bactericidal approach in the application of water disinfection. The authors propose a unique bactericidal mechanism from the mechanics' perspective using both simulation and experiments and concluded that 'tearing effect was the leading cause of bacterial destruction'. The work is of interest to the research communities that aim at developing antimicrobial-free disinfection technology to address global challenges in environment and healthcare. However, there are a few fundamental questions that need to be answered or clarified before it is accepted for publication.

Response: We appreciate the reviewer's positive comments. The questions raised by the reviewer are addressed point by point below.

1. The POPE lipid bilayer model is over-simplified for the modelling of bacterial cell wall as the cell wall structure of prokaryotic cells is very different from that of eukaryotic cells. A polysaccharide (peptidoglycan) layer is present in the bacterial cell wall which is a rigid and load-bearing component. This layer is vital to the viability of bacteria and more pronounced for the gram-positive bacteria. The simulation of bacterial cell wall mechanics should take this into consideration.

Response: We thank the reviewer for this comment. The POPE lipid bilayer is a widely-accepted simplified model for bacterial membrane, and it has been proved significant in analyzing molecular mechanism of bacteria-nanomaterial interactions in many previous studies¹²⁻¹⁴. In this work, we used the POPE model to simulate the interactions between the bacteria and the materials with different well depth parameters, and obtained the essential well depth parameter of the nanotip surface to produce a higher dispersion interaction, which provides theoretical guidance for the hydrodynamic-bactericidal mechanism.

The simulation of cell membranes with realistic components, including lipid, protein, and peptidoglycan, is the future direction of molecular dynamics simulations, but it is still a challenge at present, as it deals with complex force fields and requires enhanced sampling algorithms and substantial data processing¹⁵. Considering Gram-positive bacteria lack the outer membrane but possess a much thicker cell wall at the outermost layer, using a peptidoglycan model to reflect the behavior of Gram-positive bacteria is worthwhile. Due to the limitation in

computational power, we have collected the preliminary result of how a peptidoglycan layer interacts with our sample material in MD simulation [Redacted].

[Redacted].

[Redacted]. This simulation results help to explain the molecular mechanism for the effectiveness of this method to different types of bacteria. Discussion about the necessity of building more complex cell membrane model has been included in the revised manuscript (page 4, line 118–125).

2. The physiochemical properties of carbon coated nanowires should be fully characterised, especially their chemistry, surface charge, wetting and their contribution to the London dispersion force. It is unclear how the amorphous carbon layer on the nanowire surface changed the magnitude of the London dispersion force and why it had a strong attraction to the bacteria, leading to the increased bacterial adhesion force.

Response: The chemistry of the carbon-coated nanowires has been examined by X-ray photoelectron spectroscopy (XPS) and Raman spectroscopy. Surface charge is an important material characteristic, yet it is not the cause for London dispersion interaction. Wettability of the sample materials was examined according to the reviewer's suggestion. The contribution of amorphous carbon layer to the London dispersion interaction has been analyzed by molecular dynamics (MD) simulation. The related results are listed below for your information.

The XPS spectra (Supplementary Fig. 4) indicates that the modified NWs were composed of three elements including Cu, C and O. The C1s spectra show that the surface carbon is mainly

sp²-structured, which is consistent with a graphitic D-band at about 1580 cm⁻¹ in Raman spectra (Fig. 2d). Because the surface carbon is mainly sp²-structured carbon, the effect of surface charge can be ignored. The wettability of the Cu(OH)₂ NWs and the modified NWs was investigated using the water contact angle measurements. As is shown in Supplementary Fig. 3, the surface of the Cu(OH)₂ NWs showed super hydrophilic characteristic as the drop was sucked into the porous material as long as touching the surface. By contrast, the water droplet took about three seconds to be sucked into the modified NWs, indicating the surface hydrophilicity was slightly decreased after carbon coating treatment.

Supplementary Fig. 3 | Water contact angle measurements. a, Modified NWs. b, Cu(OH)₂ NWs.

To understand the underlying molecular mechanism behind the attractive interaction between the amorphous carbon layer and the bacterial membrane, MD simulations have been performed (Fig. 1e-g). According to the chemical composition of the carbon layer, parameters of sp² carbon atoms based on the INTERFACE force field were assigned to the amorphous

carbon atoms in the model¹⁶. The simulation results indicate that the amorphous carbon material with a well depth (ϵ) of 0.293 kJ mol⁻¹ strongly interacted with the lipid molecules and was inserted into the membrane. This strong attraction arises from the strong dispersion interaction between the sp² carbon atoms and lipid molecules, as can be seen from the sharply decreased van der Waals interaction energy upon insertion (Fig. 1f, $\epsilon = 0.293$ kJ mol⁻¹). We hope this clarification can help the reviewer to understand the interaction between the amorphous carbon layer and the bacterial membrane.

3. It is necessary to check whether the Cu(OH)₂ nanowires were unchanged or converted to something else e.g. CuO or Cu₂O after the high-temperature pyrolysis of coated glucose at 550C for 2h in Ar to form carbon coating. It is important to have a detailed information of both as-synthesised Cu(OH)₂ nanowires and glucose after heat treatment to fully understand the physiochemical characteristics of coated nanowires because they are crucial in elucidating adhesion and bactericidal mechanisms.

Response: Thanks for the valuable comment. We have revised several sentences in the Methods section to make the synthesis process more clearly understood. The Cu(OH)₂ NWs can be converted to CuO or Cu₂O at high temperatures and the nanowires will become fragile with wrinkles on their surfaces¹⁷. To prevent the undesirable change in nanowire morphology, we placed the glucose at the central heating zone while the Cu(OH)₂ NWs downwind of the tube where the temperature was no more than 100 °C (Fig. R4).

Fig. R4. Synthesis strategy of the carbon-coated Cu(OH)₂ NWs

The X-ray diffraction measurement was applied to investigate the crystalline structure of the carbon-coated Cu(OH)₂ NWs (modified NWs). As is shown in the XRD pattern of the modified NWs, three diffraction peaks located at 43°, 50°, and 74° marked with asterisks

correspond to the copper substrate, while the remaining peaks are well indexed to the orthorhombic phase of $\text{Cu}(\text{OH})_2$ (JCPDS 80-0656). No CuO or Cu_2O phase was detected. Therefore, we confirmed that the $\text{Cu}(\text{OH})_2$ phase was maintained after carbon coating and our strategy is effective to preserve the structure of the nanowires. Similar discussion has been included in the revised manuscript (page 5, line 143–145).

Supplementary Fig. 2 | XRD patterns of the $\text{Cu}(\text{OH})_2$ NWs and the modified NWs.

4. Given widely reported studies of the role of reactive oxygen species (ROS) induced by carbon-based nanomaterials (nanotubes, graphene nanosheets, quantum dots), it is suggested to measure the ROS level on the C-NW surfaces, so that other bactericidal mechanisms than mechano-killing could be ruled out.

Response: According to the reviewer's suggestion, we have measured the level of reactive oxygen species (ROS) in the bacteria treated by the modified NWs. The untreated *E. coli* sample was used as a negative control, while the Rosup reagent, provided by the supplier, was used to stimulate the bacteria to generate intracellular ROS as a positive control (see Supplementary Methods). As is shown in Supplementary Fig. 15, the ROS production was not elevated in the bacteria treated by the modified NWs, compared to the negative and positive control. Therefore, the contribution of oxidative stress in our system is negligible. Relevant discussion about ROS

has been added in the revised manuscript (page 7, line 225–227).

Supplementary Fig. 15 | Measurement of the intracellular ROS in the untreated *E. coli* cells and the *E. coli* cells treated by the modified NWs. The *E. coli* sample treated by Rosup was used as a positive control.

5. Explain why the water disinfection experiments were conducted under visible light illumination. What would be the implication if the carbon coated NWs were photocatalytic? CuO and Cu₂O are known photocatalysts that can generate ROS. If combined with carbon, they may even form heterojunctions which are likely to have excellent photocatalytic properties under visible light.

Response: During disinfection, the materials were placed inside the chamber unit which was protected from light (Fig. 4d and Supplement Fig. 10). Therefore, the contribution of photocatalytic activities can be neglected in our system. Besides, the fabrication condition of the carbon-coated Cu(OH)₂ NWs was carefully controlled to prevent the conversion of Cu(OH)₂ to CuO or Cu₂O (See response to comment 3). Nevertheless, we thank the reviewer for raising an interesting research topic on the combined photocatalytic effect of CuO or Cu₂O with carbon, which is worth exploring in the future.

After the disinfection procedure was completed, we collected the treated water and put it under visible light illumination during storage. This storage procedure is well-adopted in water treatment engineering to evaluate the safety of the disinfected water. Because after the conventional disinfection process (e.g. UV radiation), the bacteria have the potential to

reactivate under visible light illumination¹⁸. Our results show that the treated bacteria by the modified NWs would not reactivate due to severe mechanical damage, which ensures the safety of the disinfected water during subsequent use.

6. Did Fig.4e also imply other possible killing mechanisms apart from the mechano-bactericidal mechanism? If not, how the nano-feature size affects the bacterial killing?

Response: Fig. 4e demonstrates that the bactericidal performance of three different nanotips was increased by at least three orders of magnitude after carbon coating compared with their original forms, which indicates the London dispersion interaction is the key reason of the rapid inactivation of bacteria in water despite the differences in their geometry and chemical composition.

On the other hand, Fig. 4e also shows the difference in the bactericidal performance among the carbon-coated nanomaterials, indicating that after achieving a higher London dispersion interaction at the contact surface, the nanostructure geometry still has influence on the forces exerted to the bacteria, which is an interesting question and need to be further studied. To give more information about the influence of geometry, we have counted the geometrical parameters of the four kinds of nanotip arrays with significant geometric differences (Supplementary Table 6), and provided additional discussions in the revised manuscript (page 10, line 380–389; page 11, line 397–404).

Specifically, the carbon-coated ZnO nanorods showed lower bactericidal efficiency than the carbon-coated Co, Mn-LDH nanoneedles and titanate nanowires. This is because ZnO nanorods had larger diameters and a lower tip density. Typically, nanostructures with a blunt feature are supposed to deliver less mechanical stress¹, due to an enlarged contact area. Besides, a lower tip density of ZnO nanorods also reduces the possibility of bacterial contact with the nanotips during flow, which negatively impacts their bactericidal performance.

In this work, we have demonstrated a unique hydrodynamic tearing mechanism of bacteria during flow. From this regard, the tip diameter and curvature that determine the adhesion force at cell-nanotip contact surface are important geometrical parameters governing the bactericidal properties of the surface. Exploring the influence of geometrical parameters in this new force

model is a meaningful task. However, the fine adjustment of geometrical parameters of nanostructures is currently a challenging problem, especially on porous substrates. We believe the advancement in nanofabrication techniques such as reactive ion etching or deep UV lithography may help to provide more precise analysis on the role of nanostructure geometry, which will be carried out in the future.

7. Explain how gram-positive bacteria with spherical shape were teared. Any evidence to support it? As the NWs on 3D porous Cu substrates are not aligned to a specific direction, from Fig.2n, piercing rather than tearing seemed to be a more likely mechanism.

Response: We appreciate the reviewer's valuable questions. We have examined the morphological change of Gram-positive *S. aureus* before and after treatment using the SEM/TEM and observed similar mechanical damage as the *E. coli*. As seen from the SEM images, the initial *S. aureus* cells had intact cell membranes; however, after being treated by the modified NWs, the cell membranes were disrupted with holes on the surface (Supplementary Fig. 20a,b). The TEM images also indicate that the cell envelope structure was damaged (Supplementary Fig. 20c,d). However, there was only a minor leakage of cytoplasm, suggesting that Gram-positive *S. aureus* is more resistant to mechanical damage compared with Gram-negative *E. coli*, due to a thicker peptidoglycan cell wall⁵.

Supplementary Fig. 20 | Morphological change of *S. aureus*. a,b SEM images of the initial *S. aureus* (a) and *S. aureus* treated by the modified NWs (b). c,d, TEM images of the initial *S. aureus* (c) and *S. aureus* treated by modified NWs (d). (Scale bar = 200 nm)

We further investigated the forces exerted on the *S. aureus* cell during the collision and tearing processes. Because *S. aureus* is a coccus, we assume the cell envelope of the *S. aureus* as a spherical shell. [Redacted]. The stress distribution profiles of the *S. aureus* under a typical contact form were obtained by the finite element simulation [Redacted]. [Redacted].

[Redacted].

The reviewer raised a question that piercing seemed to be a more potential cause for the killing of bacteria. This is an intuition that we believe many of our readers would have without deeply exploring the force model in flowing water. As can be seen, the uncoated $\text{Cu}(\text{OH})_2$ NWs exhibited very weak bactericidal efficiency during flow and no obvious bacterial membrane damage was observed (Fig. 2f-h), which suggests when bacteria dynamically collide with a nanotip during flow, the collision energy fails to puncture or pierce a bacterium. This is confirmed by the stress distribution profiles of the bacteria obtained by finite element simulations, indicating that the maximum stresses in the collision process are two orders of magnitude lower than the critical stress for rupture (Fig. 3e,f).

By contrast, the modified NWs showed largely increased bactericidal performance. In this case, the strong attractive van der Waals interaction (London dispersion interaction) between the amorphous carbon layer and the bacteria leads to a tearing effect on nanotips during flow. We demonstrated that the tearing stress exceeds the critical stress and can rupture the bacteria (Fig. 3g,h). The numeric simulation results are in good agreement with the experimental results of $\text{Cu}(\text{OH})_2$ NWs and modified NWs, which allow us to demonstrate that the tearing generated

by the hydrodynamic and dispersion forces is the true reason for the cell rupture rather than collisions.

Reviewer #3:

*Peng et al. report a copper-based nanostructured material that is highly effective at inactivating bacteria through a hydrodynamic tearing mechanism. The authors use modeling and AFM measurements to design nanostructures that incorporate a London dispersion interaction. They construct carbon-based nanostructures within a porous copper foam and show that it removes bacteria from liquid culture by over 6 log-fold, with no trace of living bacteria detected. Using SEM and TEM, they find that bacteria have a significant amount of their membranes removed and that a significant amount bacterial debris is left on the nanostructures. The authors show significant propidium iodide fluorescence within the bacterial cytoplasm following treatment. By estimating flow velocities, cell wall rigidity, and interaction forces with the nanostructures, the authors propose that bacteria are torn apart by nanostructures rather than disrupted mechanically by collision. Finally, they show that the mechanism works against other bacteria including *P. aeruginosa* and *S. aureus*, and that nanostructures using other materials including ZnO, Co, and Titanate are effective at removing bacteria.*

Overall, I find the potential result of high-performing microbial removal to be a very exciting result that could advance the field of antimicrobial nanostructure surfaces in a significant way. The authors should also be lauded for their efforts that use different mixed technical approaches including AFM, biophysical modeling, and materials synthesis. However, I have major concerns about the microbiological characterization, lack of details throughout, and the inconsistency of the antimicrobial activity data. In addition, the data suggest contradictory rupturing mechanisms and the central conclusion of the manuscript rests on the assumption that bacteria are killed by the nanostructures rather than being adsorbed by them. Sufficient quantitative data that rules out adsorption is not provided.

Response: We thank the reviewer for the positive comments. The concerns and issues are addressed point by point below.

Major concerns:

1. Fundamental details of the bacterial growth and killing assays are lacking, which make the experiments difficult to interpret and impossible to reproduce. Examples include but are not

limited to a lack of description of the growth medium that was used to culture the bacteria, the media that was used to count CFUs, how long bacteria remained resuspended in DI water following growth, what volume of culture was treated, and over what time interval the treatment was performed. A cell density of 10^9 – 10^{10} CFU/mL is listed as a description of exponential phase. However, this is not a qualifier of exponential growth.

Response: We appreciate the reviewer’s valuable comments. The Methods section has been revised by providing a more detailed description about the bacterial growth and disinfection assays, and is listed below for your information.

(1) Culture of bacteria and preparation of bacterial suspension

The bacteria were cultured in nutrient broth at 37 °C with shaking at 150 rpm for 12 h to achieve a concentration of 10^9 – 10^{10} CFU mL⁻¹. The composition of the culture media used in this study is shown in Supplementary Table 7. The cultured bacteria were harvested by centrifugation, washed twice with sterile DI water to remove the culture medium and resuspended in DI water. Then, 1 mL of the pure bacterial culture was added to 1-L sterile DI water to achieve a concentration of 10^6 – 10^7 CFU mL⁻¹, which is typically the highest bacterial concentration detected in untreated wastewater¹⁹, and is commonly adopted to evaluate the treatment capacity of a disinfection process²⁰⁻²².

Supplementary Table 7 | Composition of the culture media

Type	Ingredient	Concentration (g L ⁻¹)	pH
Nutrient broth	Peptone	10.0	7.2 ± 0.2
	Beef extract	3.0	
	Sodium chloride	5.0	
Nutrient agar	Peptone	10.0	7.3 ± 0.1
	Beef extract	3.0	
	Sodium chloride	5.0	
	Agar	15.0	

To address the reviewer's concern about the growth phase of the *E. coli*, we measured the optical density of the bacterial culture at a wavelength of 600 nm. The optical density shows the degree of light scattering caused by the bacteria, which is correlated with the cell concentration and is routinely used to identify the stage of growth in liquid culture²³. The OD₆₀₀ measurements were recorded at different culture times using a spectrophotometer with an optical pathlength of 1 cm (SHIMADZU, UV-2600). The growth curve of *E. coli* was shown in Fig. R5, which indicates the *E. coli* was in a late exponential phase after 12-h incubation.

Fig. R5. Growth curve of *E. coli*

(2) Bactericidal tests

We conducted the bactericidal tests immediately after the diluted bacterial suspension (10^6 – 10^7 CFU mL⁻¹) was prepared. Before that, the copper foam had been washed by pure water in the flow-through device for two hours to remove surface impurities. The fresh bacterial solution was then flowed into the device. Normally, the disinfection performance tests were completed within one hour, during which the effluent water samples were collected. The flow rate was fixed at 2.7 mL min⁻¹, therefore, about 160-mL water was treated for each test. For each type of material, three sets of flow-through devices were set up and operated independently. The bactericidal efficiency for each type of material was counted based on the three replicates.

(3) Evaluation of inactivation efficiency

The live bacterial concentrations in the influent and effluent water samples were measured

using a standard plate count method. The principle of this method is that when material containing bacteria is cultured, every viable bacterium develops into a visible colony on a nutrient agar medium. The number of colonies, thus, is the same as the number of the bacteria present in the sample. For the effective result, the dilution of original sample must be performed so that less than 300 colonies of the target bacterium are grown. More than 300 colonies often lead to overlapping and errors in counting.

Specifically, 1 mL of each water sample was first collected in a sterile centrifuge tube. Next, 100 μ L of the sample was added to another sterile centrifuge tube containing 900 μ L sterile DI water to dilute tenfold. This was repeated four times to prepare serial dilutions (1:10, 1:100, 1:1000 and 1:10000). Once diluted, 100 μ L of the dilution sample from various dilutions was spread onto a sterile Petri plate (in triplicate for each dilution) to which molten and cooled nutrient agar medium was added. Following 24-h incubation at 37 °C, the number of the bacterial colonies developed on the plates can be counted, and the concentration of bacteria in the original water sample was obtained by multiplying the number of colonies obtained per plate by the dilution factor.

The inactivation performance was evaluated by logarithmic removal efficiency, which was defined by $-\log(C/C_0)$, where C and C_0 (unit: CFU mL⁻¹) represent bacterial concentrations in the influent and effluent water obtained by plate counting. When no colonies were formed on the plates, including original and diluted water samples, the bacteria in the water sample were considered as completely inactivated and the logarithmic removal efficiency was calculated by $\log(C_0)$.

2. Mechanism of bacterial rupturing. The fluorescence data and SEM/TEM data are in seeming contradiction with each other. The authors observe significant propidium iodide using epifluorescence and structured illumination (Fig. 2h and Supp. Fig. 9), which support their claim that the bacterial membrane is ruptured. Propidium iodide stains nucleic acids in the cytoplasm. However, the SEM and TEM show that cells are ruptured with defects on the order of 100-200 nm (Fig. 2j and 2l), for which the authors indicate the cells have “cytoplasm missing” (line 187). The two interpretations are contradictory because the cytoplasm cannot

both be missing due to large ruptures and be stained with propidium iodide. Given that cells are resuspended in pure water, pores of this size would result in the loss of nucleic acids and cytoplasmic content. An analysis that determines the fraction of cells that are propidium iodide positive or have loss of cytoplasm is necessary to further support the claim. Data quantifying the distribution of rupture sizes would also be helpful but are given.

Response: The reviewer raised an important concern that large rupture of the bacteria would influence the staining of nucleic acids within the bacterial cell. To address the reviewer's concern, we measured the rupture sizes of the bacteria and then examined whether cytoplasm leakage would influence the intake of propidium iodide in the damaged bacteria based on experimental results and literature review.

Supplementary Fig. 11 | Distribution of the rupture sizes of the *E. coli* cells treated by the modified NWs

The rupture sizes of the bacteria are measured from the SEM image of the bacterial sample after treatment and are presented in Supplementary Fig. 11, which indicates the average rupture size of the treated bacteria is 122 ± 32 nm (mean \pm SD from 25 different cells). However, the rupture of bacteria does not necessarily mean the cytoplasm is missing. As can be seen from the TEM image in Fig. 21, although the bacterial cell was broken with partial leakage of cytoplasm, the cell shape was maintained and the majority of the cytoplasm remained. To make this clearer, we revised the description about Fig. 21 as “*the cell envelope was ruptured, leading to partial leakage of the cytoplasmic contents*”.

Next, we would like to discuss whether partial leakage of cellular content would negate

the staining of propidium iodide. It has been reported that conventional disinfection techniques (e.g. chlorine and ultraviolet radiation) resulted in bacterial membrane damage and structure destruction, leading to leakage of cellular content (including nucleic acids)^{24,25}. Yet previous studies have shown that the damaged bacteria with partial cytoplasm missing can still be labeled by propidium iodide²⁶⁻²⁸, which suggests that the nucleic acids still existed within the bacteria. The nucleic acids can be largely missing only when the bacterial cells lyse and completely lose their cell structure. For instance, Huo *et al* studied the storage of electroporated bacterial cells in DI water²⁹, and found that the propidium iodide signals first increased due to intensified membrane damage (0–4 h). After 24-h storage, many damaged cells were disintegrated completely (Fig. R6), and at this point, the decrease of propidium iodide signals was observed.

Fig. R6 Fluorescence microscope images of PI-stained *E. coli* for different storage times and corresponding SEM images (Adapted from Huo *et al*' s work²⁹).

In our system, we prepared the bacterial samples for SEM/TEM and fluorescence microscope immediately after disinfection. At this point of just being treated, the bacteria were ruptured with partial leakage of cellular contents but the majority of the cytoplasm remained (Fig. 2l). Therefore, the nucleic acids in the cytoplasm can still bind with propidium iodide to show red fluorescence (See the bacteria treated by modified NWs in Fig. 2h). We hope this clarification could help the reviewer to understand.

3. The effect of flow on bacterial tearing seems contradictory. The model presented in Fig. 3 shows most of the rupturing force is due to the London dispersion interaction and that flow-induced collision produces a negligible amount of pressure on the bacterial membrane (2.6E-

4 MPa / 699E-4 MPa or 0.3% in the tip interaction in Fig. 3e vs. 3g). This would suggest that flow is not necessary for bacterial ruptures. However, the authors show that the modified nanostructures are largely ineffective without flow (Suppl. Fig. 12), raising an apparent contradiction.

Response: The maximum tearing stresses in the tearing process were calculated to be 6.99×10^{-2} and 5.28×10^{-2} MPa, which exceed the rupture stress (0.05 MPa) of the bacteria. Therefore, the bacterial cell can be ruptured by the tearing stress produced by hydrodynamic and dispersion forces. A detailed description about the stress distribution profiles in the collision and tearing processes is listed below.

Fig. 3 shows the stress exerted on the bacterial cell under two types of interaction. During water flow induced cell collisions, the maximum stresses were calculated to be 2.61×10^{-4} and 4.54×10^{-4} MPa, respectively (Fig. 3e,f), which are two orders of magnitude lower than the critical value (0.05 MPa). Such data suggests that the collision process cannot mechanically rupture a bacterium, and explains the poor bactericidal performance of the surface of the $\text{Cu}(\text{OH})_2$ NWs. During the tearing process induced by the coupling of water flow and the London dispersion interaction, the maximum tearing stresses were calculated to be 6.99×10^{-2} and 5.28×10^{-2} MPa, respectively (Fig. 3g,h), which exceed the rupture stress of the bacteria (0.05 MPa).

Based on the above discussion, we can conclude that the bacteria were torn up by the coupling of hydrodynamic force and the higher dispersion interaction between the nanotip surface and the cell envelope. Thus, the flow is a critical factor for bacterial rupture, which is consistency with the observation that the modified NWs were largely ineffective without flow (Supplementary Fig. 19). Please note that the content of Supplementary Fig. 12 has been placed in Supplementary Fig. 19 in the revised version.

4. Bacteria are cultured to a density of 10^9 - 10^{10} CFU/mL, which suggests growth in a rich medium. Subsequent resuspension into pure water could sensitize bacteria a number of non-physiological ways including nutrient and osmolarity shocks. Is the bacterial removal effect by nanostructures observed if bacteria are cultured in a low osmolarity / low nutrient environment

or introduced into the device using the same growth medium instead of pure water?

Response: We have tested the disinfection performance of the modified NWs in real water samples including the tap water and reclaimed water (Fig. 4c). As shown in Supplementary Table 5, the two water samples exhibited low levels of conductivity and organic matters, corresponding to a low osmolarity and low nutrient environment. A rapid decrease of live bacterial concentration was observed in both water samples after treatment.

We used the nutrient-rich medium to culture the bacteria in order to obtain a sufficiently large density of bacteria cell for laboratory tests. We did not introduce the bacterial culture with growth medium into the device, because the culture medium contains high concentrations of nutrient and mineral salts, which is substantially different from the condition in raw water or surface water to be treated.

5. The central conclusion of the paper rests on the assumption that cells are killed by the nanostructures. An alternative explanation for the results is that bacteria are largely adsorbed to the nanostructures but this scenario is not adequately addressed. The authors state that 360-400 cells per image was observed in the influent and effluent (lines 177-199). However, this spot analysis is not sufficient to support the conclusion, as the cell density in microscopy images is highly variable and does not provide adequate statistical sampling. The authors should provide an appropriate sampling analysis to quantify the number of cells in the influent and effluent in all treatments, which should range in the 10^6 - 10^7 CFU/ml.

Response: We appreciate the reviewer's insightful and constructive comment. We believe excluding the contribution of absorption can help to strengthen the bactericidal mechanism in this work.

Optical density shows the degree of light scattering caused by the bacteria, which is correlated with the cell concentration and is widely used to reflect the density of cells in liquid culture²³. We have collected the optical density data of the influent and effluent water at a wavelength of 600 nm (OD₆₀₀) to quantify the number of bacterial cells (Supplementary Fig. 14). We first determined the OD₆₀₀ value of the *E. coli* suspension (4×10^6 CFU mL⁻¹, determined by plate count), which lies in 0.02–0.03 at an optical pathlength of 5 cm (See

Supplementary Methods). When the *E. coli* suspension was diluted tenfold, the OD₆₀₀ value dropped significantly to lower than 0.005 (Supplementary Fig. 14a), confirming that the OD₆₀₀ was sensitive to the change of cell density in water.

The OD₆₀₀ values of the effluent water at different flow rates were collected (Supplementary Fig. 14b). The potential influence of released particles from the modified NWs was excluded by using the DI water as the influent. As can be seen, the OD₆₀₀ values of the effluent DI water were near or below the detection limit, indicating the turbidity of the effluent water was not increased by the material itself. Based on this, we can conclude that the OD₆₀₀ values of the effluent *E. coli* suspension were comparable to the influent, meaning that the density of the bacterial cells was unchanged in the effluent. Thus, the bacteria treated by the modified NWs were not removed by adsorption. Relevant discussion has been included in the revised manuscript (page 6, line 222–225).

Supplementary Fig. 14 | Optical density Measurements. a, OD₆₀₀ of the *E. coli* suspension in different concentrations (***) $p < 0.001$, student T-test). b, OD₆₀₀ of the effluent water at different flow rates. The *E. coli* suspension (4×10^6 CFU mL⁻¹, determined by plate count) and the DI water were used as the influent water (# below the detection limit).

6. The removal efficacy is listed but no units are given (Fig. 2f). Are these CFUs / mL? Absolute CFUs? If so, what volumes are treated in each condition? This is data that is critical to the conclusion but insufficient information is provided to properly evaluate the effect. The full CFU data for both the influent and effluent should be provided.

Response: The logarithmic removal efficiency is defined by $-\log(C/C_0)$, where C and C_0 (unit:

CFU mL⁻¹) represent bacterial concentrations in the influent and effluent water obtained by plate counting. Therefore, the removal efficiency is dimensionless. The full CFU data for both the influent and effluent has been provided in Supplementary Table 1.

Supplementary Table 1 | Live bacterial concentrations in the treated and untreated water samples (raw data of Fig. 2f).

Concentration (CFU mL ⁻¹)	Mean	Std
Influent water	4.5×10^6	2×10^5
Effluent of the modified NWs	0	0
Effluent of the Cu(OH) ₂ NWs	2.75×10^5	2.5×10^4
Effluent of the modified Cu	4.85×10^5	3.15×10^5

Other issues:

1. The bacteria in Fig. 2k does not appear to be exponential phase E. coli and the cells do not have the same shape or aspect ratio as the cells in the SEM image (Fig. 2i). This raises a concern that the microbial growth conditions may not be consistent between experiments.

Response: The TEM image (Fig. 2k) shows cross sections of the bacterial cells. Therefore, although *E. coli* cells are rod-shaped as the SEM image shows (Fig. 2i), they can exhibit different sizes and aspect ratios at different cross sections (Fig. R7). To clarify this point, we would like to describe the differences in the sample preparation procedure for TEM and SEM. For TEM, after fixation and dehydration, the bacterial samples were infiltrated with resin. The hardened resin was then sectioned extremely thinly, at about 100–150 nm. This allows for the electron beam to pass from the electron gun through the specimen to the detector. By contrast, for SEM, after fixation and dehydration, the bacterial samples were dried and directly observed using SEM after sputter-coat.

We have rewritten the Methods section with more details about the culture conditions of the bacteria. Specifically, the bacteria were cultured with nutrient broth medium at 37 °C on a shaker (150 rpm) for 12 h to achieve a concentration of 10⁹–10¹⁰ CFU mL⁻¹. We kept the same culture time in different experiments to ensure the growth phase of the bacteria was identical.

The growth curve of the bacteria has been provided in the response for *major concern 1*, which indicates the *E. coli* was in a late exponential phase after 12-h incubation (Fig. R5).

Fig. R7 TEM image of the *E. coli*

2. It is unclear to what extent pores are present in Supp. Fig. 9 and how representative the structured illumination image is. The sample size is not large enough to give statistical significance and it is unclear whether the sections of the DIO fluorescent image indicate a pore is present, are due to irregular staining, or due to adjustment of the lookup tables. In addition, one of the treated cells is not rod-shaped, which does not match the EM images in 2j and 2l or the claim that the cell is from exponential phase.

Response: The structural illumination microscopy (SIM) surpasses the diffraction limit of conventional optical microscopy by using patterned illumination, which generates a super resolution of ~ 100 nm³⁰. As SIM technology has been proved useful in producing high-resolution images of eukaryotic cells, we were interested if it could be used to directly observe the damage of the cell envelope. We conducted SIM imaging to evaluate the morphology of *E. coli* cell before and after treatment (Supplementary Fig. 12, corresponding to Supplementary Fig. 9 in the original version). As can be seen, the *E. coli* cell treated by modified NWs had a compromised membrane, and the entrance of red propidium iodide confirmed the membrane damage. However, we have to point out that SIM did not provide more precise information about the cell envelope structure as we expected, due to a small size of bacteria (~ 1 μ m). Nevertheless, the result of SIM imaging was consistent with the observations using fluorescence microscopy and electron microscopy, and was provided as supplementary

information.

The reason why the bacteria cells under SIM imaging appeared different shapes compared with the SEM image is that the SIM captures the optical section of the bacteria³¹. By contrast, the SEM detects the secondary electrons which contain topographic information of the sample³². The cultural time of bacteria was kept the same (12 h) to ensure the growth phase of the bacteria was identical in different experiments. The growth curve of the bacteria has been provided in the response for *major concern 1*, which indicates the *E. coli* was in a late exponential phase after 12-h incubation (Fig. R5).

3. The resolution of the epifluorescence images in Fig. 2h is insufficient to make a conclusion. The data should be quantified to determine which fraction of cells in the overall effluent is positive for propidium iodide.

Response: The live/dead assay coupled with fluorescence microscope is widely performed to evaluate bacterial viability³³. Although this technique cannot give absolutely quantitative data, it can easily distinguish bacterial cells that are highly viable and dead cells with severely damaged membrane^{2,12,34,35}.

In our experiment, we filtered 1-mL effluent sample through a black polycarbonate membrane for observation (see Methods), which prevented the vibration caused by the movement of bacteria in water. The bacteria in the control groups remained alive as indicated by the lack of red fluorescence (Fig. 4h). Only the modified NWs resulted in the intensification of the red fluorescence signal, and there were over 99% of the bacteria stained positive for propidium iodide. This fraction was calculated by dividing the number of red cells with the total number of green and red cells^{36,37}. The staining results indicated that the bacteria treated by modified NWs were severely damaged and lost viability, which was in good agreement with the quantitative plate counting results (Fig. 2f).

4. Given a rate of 2.7 mL/min, it would take approximately 12 hours to filter the ~2L bacterial resuspension in their setup (Supp. Fig. 8). During this time, E. coli in pure water may cease to swim due to the lack of nutrients. However, it does not appear that the influent is homogenized

by any type of stirring mechanism. Are all experiments (modified NW, modified Cu, Cu(OH)₂ NW) performed over the same time period with homogenization, using the same volume, and same rate of flow for all devices? Information about these conditions are not given.

Response: We appreciate the reviewer's comment. These experimental details are crucial to make the results repeatable, which have been added into Methods section in the revised manuscript. In fact, the influent water was kept homogenized by a magnetic stirrer, which can be seen in Supplementary Fig. 10 (corresponding to Supplementary Fig. 8 in the original version). The experimental procedures to evaluate the disinfection performance of modified NWs, modified Cu, Cu(OH)₂ NWs were kept the same, which are listed below:

The disinfection tests were performed immediately after the diluted bacterial suspension (10^6 – 10^7 CFU mL⁻¹) was prepared. Before that, the copper foam had been washed by pure water in the flow-through device for two hours to remove surface impurities. A sterile magnetic stir bar was put into the bacterial suspension to run magnetic stirring. A peristaltic pump was used to flow the water into the device, and the flow rate was fixed at 2.7 mL min⁻¹. Normally, the disinfection performance tests were completed within one hour, during which the effluent water samples were collected. Therefore, about 160-mL water was treated for each test. For each type of material, three sets of flow-through devices were set up and operated independently to evaluate the disinfection performance. The disinfection efficiency for each type of material was counted based on the three replicates.

5. Many figure captions contain insufficient detail. For example, it is unclear from the caption alone what the velocity field boundaries are in Fig. 3C, what the numbers in red in Fig. 3e-h represent, and what the 0.05 on the nanotip represents.

Response: We appreciate the reviewer's comment. We have checked the figure captions and rewritten them with more detail. In Fig. 3c, a right triangle domain was selected for computation, in which the hypotenuse was defined as the inlet boundary, while the other two sides were defined as the outlet boundaries. In Fig. 3e-h, the maximum stresses exerted at each contact form are denoted in red, and are compared with the critical stress (0.05 MPa).

6. A bacterial culture density of 10^6 - 10^7 CFU/mL is used in the device. Is this culture density critical? Are the nanostructures effective at higher or lower culture densities? Does the undiluted 10^9 - 10^{10} CFU/mL culture overwhelm the device?

Response: The bacterial density of 10^6 - 10^7 CFU mL⁻¹ is normally the highest concentration of faecal bacteria in untreated wastewater¹⁹. The bacterial concentrations in raw water and source water for potable water generation are lower than this value. Therefore, a bacterial suspension of 10^6 - 10^7 CFU mL⁻¹ is commonly adopted to evaluate the treatment capacity of a water disinfection process²⁰⁻²².

In this work, we demonstrated that the modified NWs can effectively disinfect water containing 10^6 - 10^7 CFU mL⁻¹ bacteria, achieving > 6 log inactivation (Fig. 2f). In the 30-d disinfection test, a diluted bacterial suspension (10^3 - 10^4 CFU mL⁻¹) was used as the influent, which is more representative for influent water in a realistic water purification plant³⁸. Complete disinfection was also achieved for a lower bacterial concentration (Fig. 4d).

This study aims at the removal of pathogenic microbes in water. As the concentration of bacteria would not reach as high as 10^9 - 10^{10} CFU mL⁻¹ either in natural water bodies or in wastewater, we did not test the inactivation performance of the device with this concentration. We hope this clarification can help the reviewer to understand the disinfection assay in this work.

7. If the London dispersion is responsible for rupturing, does this suggest that ions in solution would negate the effect? It would be helpful if the authors addressed this issue to understand potential limitations of the device.

Response: In this study, we have tested the disinfection performance of the modified NWs in real water samples including tap water and reclaimed water (Fig. 4c). The reclaimed water contained more ions than the tap water as indicated by a larger conductivity (Supplementary Table 5). The ions in the water samples were not observed to negate the bactericidal effect, as a rapid decrease of live bacterial concentration was achieved in both the tap water and reclaimed water after treatment. Nevertheless, the types and concentrations of ions vary in different water sources, and the adsorption of ions on the nanotip surface is likely to change the level of

dispersion interaction between the bacteria and nanotip surface. Exploring the impact of ions on the hydrodynamic-bactericidal mechanism is of importance for its practical applications, which is discussed in the revised manuscript (page 11, line 419–427).

Reviewer #4:

The elimination of pathogens from water has long been a major challenge. In this study, the authors investigated the hydrodynamic tearing of bacteria on nanotips for sustainable water disinfection. A rapid inactivation of >99.9999% of the bacteria in water in a 30-day field test was observed. The authors proposed that that the mild water flow can efficiently tear up the bacteria through the higher dispersion force between the nanotip surface and the cell envelope. It was further demonstrated that tearing of the cell-envelope led to the rupture of the bacteria rather than collisions.

Since the effective mechanical disinfection in bulk water has not been investigated, this manuscript provided an innovative approach for water disinfection. This manuscript provides a valuable contribution to the topic and thus merits publication after minor revisions. A few comments/concerns are shown below for the authors' consideration.

Response: We thank the reviewer for the positive comments. The concerns and issues raised by the reviewer are addressed point by point below.

1. The disinfection effectiveness was evaluated at a flux of $2 \text{ m}^3 \text{ h}^{-1} \text{ m}^{-2}$. How will the flow rate of water effect the disinfection efficacy?

Response: There are multifaced effects of flow rate on the disinfection performance of our system. The flow rate determines the magnitude of the hydrodynamic force. Therefore, increasing the flow rate means more stress can be delivered to the bacteria. The flow rate also influences the contact time of the bacteria within the materials. A higher flow rate corresponds to a lower contact time, leading to lower contact possibility between the bacteria and the nanotips³⁵.

We have tested the disinfection performance of the modified NWs at different flow rates (Supplementary Fig. 19). The bactericidal test in static condition showed limited bactericidal activity (~1.4 log), indicating that the significance of water flow to the rupture of the bacteria. In the flow condition, the modified NWs achieved remarkable inactivation of >99.9999% *E. coli* (>6 log removal) in water at a flux of $0.5\text{--}2 \text{ m}^3 \text{ m}^{-2} \text{ h}^{-1}$, which is the commonly-adopted flux in filtration modules for water treatment^{10,11}. Yet when the flux further increased to 6 m^3

$\text{m}^{-2} \text{h}^{-1}$, a decrease in the inactivation efficiency was observed, which suggests that the negative impact of a lower contact time at such a high flux is more significant compared with the impact of an increased stress. Similar discussion about the influence of flow rate has been added in the revised manuscript (page 9, line 323–332).

Supplementary Fig. 19 | Influence of different flow rates on the inactivation of bacteria by the modified NWs. The flow rate is given in the form of flux ($\text{m}^3 \text{h}^{-1} \text{m}^{-2}$). Flow rate of zero represents the bactericidal test of the modified NWs in the static condition.

2. As the authors have mentioned the energy consumption of conventional disinfection practices (chemicals, UV, etc). Can the authors provide some insights on the comparison of hydrodynamic tearing of bacteria on nanotips with conventional disinfection practices in terms of energy consumption. This is extremely important with regard to its practical application.

Response: In this study, we reported a methodology on exploiting fluidic energy to destroy pathogenic bacteria. A superior inactivation of $>99.9999\%$ bacteria was achieved by simply flowing the water through the device. That is a process in which, besides the nanotip surface, flow of contaminated water is the only requirement to attain disinfection, which avoids toxic chemical byproducts and additional energy input. Similar discussion has been added in the revised manuscript (page 12, line 432–434). A comparison between the conventional disinfection techniques and our method has been provided in Supplementary Discussion 4.

3. For the long-term disinfection experiment, diluted E. coli stock solution was applied as the

feed water. However, colloids, particles, dissolved organic matter, and ions coexist with pathogens in realistic water treatment conditions. Will these co-existing substances influence the interaction between pathogens and the modified NWs?

Response: As the reviewer pointed out, there are colloids, particles, dissolved organic matter and ions existing in realistic water samples. These substances may adsorb to the modified NWs, shield the effective sites on the nanotip surface, and possibly change the level of dispersion interaction with the bacteria, thus negatively influencing the disinfection performance of the device. For practical water treatment, the modified NWs can be combined with other conventional water treatment process. For example, the influent water can be pretreated by an ultrafiltration module to remove most of the impurities³⁹.

4. Did the author investigate the potential leakage of nanotips during operation?

Response: The leakage of nanowires was investigated by measuring the copper concentration in the effluent, which was in a low concentration (0.3–0.5 mg L⁻¹) compared with the guideline threshold of drinking water (2 mg L⁻¹)¹⁹. Besides, the morphology of the modified NWs was examined after 30-day operation, which indicates that the majority of the nanowires remained intact (Supplementary Fig. 21). Based on these observations, the leakage of nanotips was limited during operation.

5. Line 209-219. As the disinfection mechanisms are different among metal nanoparticles, nanostructured surfaces, and the modified NWs, is the different contact time required for effective disinfection?

Response: The mechano-bactericidal actions have been discovered at surfaces of different nanomaterials, including nanostructured surfaces and metal nanoparticles³. Typically, previously reported mechano-bactericidal actions are based on a surface-contact mechanism, in which bacterial cells are deformed during static contact with the sharp nanostructures. This mechanism mostly requires a long contact time (up to hours) to deliver sufficient stress beyond the elastic limit of the cell envelope (Fig. 2o), causing membrane disruption and subsequent cell death^{1,4,40,41}.

In this work, we demonstrated that dispersion interactions between the contact surface and bacteria play a key role in transforming the kinetic energy from water flow to destroy the bacteria. With this principle, we designed a nanostructured surface with a higher dispersion interaction, at which the water flow can tear up the bacteria during instantaneous contact with the nanotips. By synthesizing a model nanotip surface on a porous three-dimensional substrate (modified NWs), we achieved effective mechanical inactivation within a short contact time (e.g. 7 s).

6. Fig. 2 (o). For carbon nanomaterials, they required long contact time while the log removal values were relatively low. Which factor is responsible for the different performances of carbon nanomaterials and modified NWs with carbon coating in this study?

Response: The bactericidal mechanism of the reported carbon nanomaterials, represented by graphene nanosheets and carbon nanotubes, is different from our system. They inactivate bacteria through a surface-contact mechanism in which continuous mechanical stress can be delivered followed by the increase of oxidative stress, therefore the inactivation process is time-dependent. For instance, the one-atom thick graphene nanosheets or sharp carbon nanotubes can cut or piece into the cell membrane^{40,42}. Super high aspect ratio carbon nanotubes can also deform bacteria through release of bending energy. This physical disruption does not rupture the bacteria, but leads to an increased level of intracellular reactive oxygen species, causing gradual decrease of cell viability².

In this work, the amorphous carbon layer was used to achieve a higher London dispersion interaction between the bacteria and nanotip surface. With this design, the fluidic energy was successfully transformed to tear up bacteria on the surface of the modified NWs. The coupling of hydrodynamic and dispersion forces is the key to the physical rupture of bacteria in our system.

7. Line 320-321, there was bacterial debris accumulated in the first unit during the 30-day operation. Will this cause the biofouling of nanotips? Will the co-existing substances (colloids, etc) in water accelerate the accumulation of cell debris?

Response: Due to the London dispersion interaction between the nanotip surface and the bacteria, bacterial debris were torn up from the bacteria cell and remaining on the tips. Yet the bacteria cannot survive on the foam surface because the nanotip pattern can exert mechano-bactericidal actions to prevent biofilm formation according to previous literature³. Nevertheless, the accumulation of bacterial debris would shield the effective sites on the nanotips, together with the co-existing substances (e.g. colloids), leading to a decrease in the disinfection performance of the device in the long term. Thus, effective regeneration method is needed to guarantee the application of this system in realistic water treatment conditions, which is one of our future research targets.

8. Line 323-325, limitation might exist here, as this calculation was based on the disinfection performance with diluted E. coli solution as the feed water.

Response: In the 30-d continuous disinfection test, a diluted bacterial suspension (10^3 – 10^4 CFU mL⁻¹) was used as the influent (Fig. 4d). This is because the bacteria present in the drinking water or advanced water treatment processes are usually in this range of concentration^{38,43,44}. To approximate the actual bacterial concentrations in practical applications, we adopted a diluted bacterial suspension to test the long-term performance of the device.

9. Line 335-339, substantially different nanostructures were observed with Cu(OH)₂, ZnO, LDH, and titanate as support layers, which would certainly influence the interaction between cell and the surfaces. More discussions may be needed.

Response: There are differences in the bactericidal performance among the carbon-coated nanomaterials, indicating that after achieving a higher London dispersion interaction at the contact surface, the nanostructure geometry still has influence on the forces exerted to the bacteria, which is an interesting question and need to be further studied. To give more information about the influence of geometry, we have counted the geometrical parameters of the four kinds of nanotip arrays with significant geometric differences (Supplementary Table 6), and provided additional discussions in the revised manuscript (page 10, line 380–389; page 11, line 397–404).

Supplementary Table 6 | Geometrical parameters of the nanomaterials used in this study

Nanomaterial	Density (30 μm^2)	Length (nm)	Diameter (nm)	Aspect ratio
Cu(OH) ₂	165	5326 \pm 890	199 \pm 32	26.6
ZnO	135	2123 \pm 539	340 \pm 70	6.2
Co, Mn-LDH	195	2883 \pm 700	127 \pm 20	22.7
Titanate	175	2911 \pm 617	47 \pm 8	61.9

Geometrical data were collected from at least 30 individual nanotips in SEM images.

Specifically, the carbon-coated ZnO nanorods showed lower bactericidal efficiency than the carbon-coated Co, Mn-LDH nanoneedles and titanate nanowires. This is because ZnO nanorods had larger diameters and a lower tip density. Typically, nanostructures with a blunt feature are supposed to deliver less mechanical stress¹, due to an enlarged contact area. Besides, a lower tip density of ZnO nanorods also reduces the possibility of bacterial contact with the nanotips during flow, which negatively impacts their bactericidal performance.

In this work, we have demonstrated a unique hydrodynamic tearing mechanism of bacteria during flow. From this regard, the tip diameter and curvature that determine the adhesion force at cell-nanotip contact surface are important geometrical parameters governing the bactericidal properties of the surface. Exploring the influence of geometrical parameters in this new force model is a meaningful task. However, the fine adjustment of geometrical parameters of nanostructures is currently a challenging problem, especially on porous substrates. We believe the advancement in nanofabrication techniques such as reactive ion etching or deep UV lithography may help to provide more precise analysis on the role of nanostructure geometry, which will be carried out in the future.

10. Line 352-357, inactivation of virus with a much smaller size was not investigated in this study. It would also have been better not to mention “exploiting fluidic energy to destroy pathogens” since only tearing of bacteria was studied. To increase the broad impact of this work, the limitation for the virus inactivation may be mentioned.

Response: Viruses are waterborne pathogens of concern. Due to their largely different structure

compared with bacteria, the inactivation of viruses using the hydrodynamic tearing method needs to be further studied. According to the reviewer's suggestion, the statement mentioned by the reviewer has been edited as "*exploiting fluidic energy to destroy pathogenic bacteria*". The potential limitation of this method towards virus inactivation has been mentioned in the revised manuscript (page 11, line 414–418).

References

- 1 Michalska, M. *et al.* Tuning antimicrobial properties of biomimetic nanopatterned surfaces. *Nanoscale* **10**, 6639-6650 (2018).
- 2 Linklater, D. P. *et al.* High aspect ratio nanostructures kill bacteria via storage and release of mechanical energy. *ACS Nano* **12**, 6657-6667 (2018).
- 3 Linklater, D. P. *et al.* Mechano-bactericidal actions of nanostructured surfaces. *Nat. Rev. Microbiol.* **19**, 8-22 (2021).
- 4 Linklater, D. P. *et al.* Influence of nanoscale topology on bactericidal efficiency of black silicon surfaces. *Nanotechnology* **28**, 245301 (2017).
- 5 Jenkins, J. *et al.* Antibacterial effects of nanopillar surfaces are mediated by cell impedance, penetration and induction of oxidative stress. *Nat. Commun.* **11**, 1626 (2020).
- 6 Francius, G., Domenech, O., Mingeot-Leclercq, M. P. & Dufrêne, Y. F. Direct observation of *Staphylococcus aureus* cell wall digestion by lysostaphin. *J. Bacteriol.* **190**, 7904-7909 (2008).
- 7 Formosa-Dague, C. *et al.* Zinc-dependent mechanical properties of *Staphylococcus aureus* biofilm-forming surface protein SasG. *Proc. Natl. Acad. Sci. U.S.A.* **113**, 410-415 (2016).
- 8 Vaiwala, R., Sharma, P. & Ganapathy Ayappa, K. Differentiating interactions of antimicrobials with Gram-negative and Gram-positive bacterial cell walls using molecular dynamics simulations. *Biointerphases* **17**, 061008 (2022).
- 9 Inou, Eacute, Takao & Osatake, H. A New Drying Method of Biological Specimens for Scanning Electron Microscopy: The t-Butyl Alcohol Freeze-drying Method. *Arch. Histol. Cytol.* **51**, 53-59 (1988).
- 10 Shimizu, Y. *et al.* Filtration characteristics of hollow fiber microfiltration membranes used in membrane bioreactor for domestic wastewater treatment. *Water Res.* **30**, 2385-2392 (1996).
- 11 Kiso, Y. *et al.* Wastewater treatment performance of a filtration bio-reactor equipped with a mesh as a filter material. *Water Res.* **34**, 4143-4150 (2000).
- 12 Fang, G. *et al.* Differential Pd-nanocrystal facets demonstrate distinct antibacterial activity against Gram-positive and Gram-negative bacteria. *Nat. Commun.* **9**, 129 (2018).
- 13 Chen, Y. *et al.* Synergetic lipid extraction with oxidative damage amplifies cell-membrane-destructive stresses and enables rapid sterilization. *Angew. Chem. Int. Ed.* **60**, 7744-7751 (2021).
- 14 Li, Y. *et al.* Graphene microsheets enter cells through spontaneous membrane penetration at edge asperities and corner sites. *Proc. Natl. Acad. Sci. U.S.A.* **110**, 12295 (2013).
- 15 Marrink, S. J. *et al.* Computational modeling of realistic cell membranes. *Chem. Rev.* **119**, 6184-6226 (2019).
- 16 Heinz, H., Lin, T.-J., Kishore Mishra, R. & Emami, F. S. Thermodynamically consistent force fields for the assembly of inorganic, organic, and biological nanostructures: The

- INTERFACE force field. *Langmuir* **29**, 1754-1765 (2013).
- 17 Shi, W. *et al.* Carbon coated Cu₂O nanowires for photo-electrochemical water splitting with enhanced activity. *Appl. Surf. Sci.* **358**, 404-411 (2015).
 - 18 Li, G.-Q. *et al.* Comparison of UV-LED and low pressure UV for water disinfection: Photoreactivation and dark repair of Escherichia coli. *Water Res.* **126**, 134-143 (2017).
 - 19 WHO. Guidelines for drinking-water quality: Fourth edition incorporating the first and second addenda. Report No. CC BY-NC-SA 3.0 IGO, (World Health Organization, Geneva, 2022).
 - 20 Oguma, K. *et al.* Application of UV light emitting diodes to batch and flow-through water disinfection systems. *Desalination* **328**, 24-30 (2013).
 - 21 Vecitis, C. D. *et al.* Electrochemical multiwalled carbon nanotube filter for viral and bacterial removal and inactivation. *Environ. Sci. Technol.* **45**, 3672-3679 (2011).
 - 22 Liu, C. *et al.* Rapid water disinfection using vertically aligned MoS₂ nanofilms and visible light. *Nat. Nanotechnol.* **11**, 1098-1104 (2016).
 - 23 Myers, J. A., Curtis, B. S. & Curtis, W. R. Improving accuracy of cell and chromophore concentration measurements using optical density. *BMC Biophys.* **6**, 1-16 (2013).
 - 24 Venkobachar, C., Iyengar, L. & Rao, A. V. S. P. Mechanism of disinfection: effect of chlorine on cell membrane functions. *Water Res.* **11**, 727-729 (1977).
 - 25 Krishnamurthy, K., Tewari, J. C., Irudayaraj, J. & Demirci, A. Microscopic and spectroscopic evaluation of inactivation of Staphylococcus aureus by pulsed UV light and infrared heating. *Food Bioprocess Technol.* **3**, 93-104 (2010).
 - 26 Kim, S. *et al.* Bacterial inactivation in water, DNA strand breaking, and membrane damage induced by ultraviolet-assisted titanium dioxide photocatalysis. *Water Res.* **47**, 4403-4411 (2013).
 - 27 Zeng, X., McCarthy, D. T., Deletic, A. & Zhang, X. Silver/reduced graphene oxide hydrogel as novel bactericidal filter for point-of-use water disinfection. *Adv. Funct. Mater.* **25**, 4344-4351 (2015).
 - 28 Jia, S. *et al.* Disinfection characteristics of Pseudomonas peli, a chlorine-resistant bacterium isolated from a water supply network. *Environ. Res.* **185**, 109417 (2020).
 - 29 Huo, Z.-Y. *et al.* Nanowire-modified three-dimensional electrode enabling low-voltage electroporation for water disinfection. *Environ. Sci. Technol.* **50**, 7641-7649 (2016).
 - 30 Huang, B., Babcock, H. & Zhuang, X. Breaking the diffraction barrier: Super-resolution imaging of cells. *Cell* **143**, 1047-1058 (2010).
 - 31 Heintzmann, R. & Huser, T. Super-resolution structured illumination microscopy. *Chem. Rev.* **117**, 13890-13908 (2017).
 - 32 Goldstein, J. I. *et al.* *Scanning electron microscopy and X-ray microanalysis*. (Springer, 2017).
 - 33 Boulos, L. *et al.* LIVE/DEAD® BacLight™: application of a new rapid staining method for

- direct enumeration of viable and total bacteria in drinking water. *J. Microbiol. Methods* **37**, 77-86 (1999).
- 34 Jiang, Y. *et al.* Hydrophilic nanoparticles that kill bacteria while sparing mammalian cells reveal the antibiotic role of nanostructures. *Nat. Commun.* **13**, 197 (2022).
 - 35 Huo, Z.-Y. *et al.* Synergistic nanowire-enhanced electroporation and electrochlorination for highly efficient water disinfection. *Environ. Sci. Technol.* **56**, 10925-10934 (2022).
 - 36 Valiei, A. *et al.* Hydrophilic mechano-bactericidal nanopillars require external forces to rapidly kill bacteria. *Nano Lett.* **20**, 5720-5727 (2020).
 - 37 Liu, L. *et al.* Mechanical penetration of β -lactam-resistant Gram-negative bacteria by programmable nanowires. *Sci. Adv.* **6**, eabb9593 (2020).
 - 38 Lin, W., Yu, Z., Zhang, H. & Thompson, I. P. Diversity and dynamics of microbial communities at each step of treatment plant for potable water generation. *Water Res.* **52**, 218-230 (2014).
 - 39 Schäfer, A. I., Fane, A. G. & Waite, T. D. Fouling effects on rejection in the membrane filtration of natural waters. *Desalination* **131**, 215-224 (2000).
 - 40 Lu, X. *et al.* Enhanced antibacterial activity through the controlled alignment of graphene oxide nanosheets. *Proc. Natl. Acad. Sci. U.S.A.* **114**, E9793-E9801 (2017).
 - 41 Ivanova, E. P. *et al.* Bactericidal activity of black silicon. *Nat. Commun.* **4**, 2838 (2013).
 - 42 Kang, S., Pinault, M., Pfefferle, L. D. & Elimelech, M. Single-walled carbon nanotubes exhibit strong antimicrobial activity. *Langmuir* **23**, 8670-8673 (2007).
 - 43 Cui, Q. *et al.* Bacterial removal performance and community changes during advanced treatment process: a case study at a full-scale water reclamation plant. *Sci. Total Environ.* **705**, 135811 (2020).
 - 44 Llorens, E. *et al.* Water quality improvement in a full-scale tertiary constructed wetland: Effects on conventional and specific organic contaminants. *Sci. Total Environ.* **407**, 2517-2524 (2009).

Reviewer #1 (Remarks to the Author):

Thank you for addressing my main concerns by providing further experimental data and discussing the limitations and future outlook of this research in the revised version of the manuscript.

Reviewer #2 (Remarks to the Author):

[No comments for authors]

Reviewer #3 (Remarks to the Author):

The authors have addressed my concerns.

Reviewer #4 (Remarks to the Author):

The authors have addressed my concerns and it can be publishable now.